# DREAMS: Preserving both Local and Global Structure in Dimensionality Reduction

**Noël Kury**    *noel-elia.kury@student.uni-tuebingen.de*
*Hertie Institute for AI in Brain Health*
*University of Tübingen, Germany*

**Dmitry Kobak**    *dmitry.kobak@uni-tuebingen.de*
*Hertie Institute for AI in Brain Health*
*University of Tübingen, Germany*

**Sebastian Damrich**    *sebastian.damrich@uni-tuebingen.de*
*Hertie Institute for AI in Brain Health*
*University of Tübingen, Germany*

**Reviewed on OpenReview:** *https://openreview.net/forum?id=xpGu3Sichc*

## Abstract

Dimensionality reduction techniques are widely used for visualizing high-dimensional data in two dimensions. Existing methods are typically designed to preserve either local (e.g., *t*-SNE, UMAP) or global (e.g., MDS, PCA) structure of the data, but none of the established methods can represent both aspects well. In this paper, we present DREAMS (Dimensionality Reduction Enhanced Across Multiple Scales), a method that combines the local structure preservation of *t*-SNE with the global structure preservation of PCA via a simple regularization term. Our approach generates a spectrum of embeddings between the locally well-structured *t*-SNE embedding and the globally well-structured PCA embedding, efficiently balancing both local and global structure preservation. We benchmark DREAMS across eleven real-world datasets, showcasing qualitatively and quantitatively its superior ability to preserve structure across multiple scales compared to previous approaches.

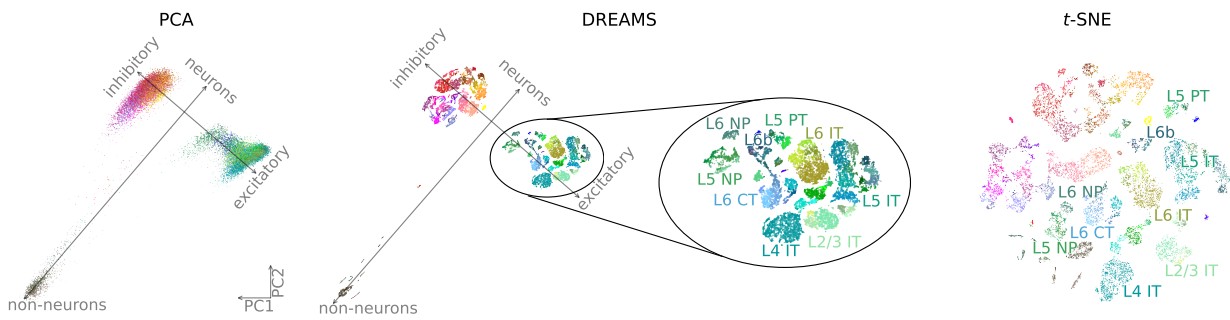

Figure 1: PCA, DREAMS, and *t*-SNE embeddings of the Tasic et al. dataset illustrate how DREAMS preserves the global organization seen in the PCA embedding — such as the separation of non-neurons, inhibitory, and excitatory neurons — while also capturing the local cell-type structure that is present in the *t*-SNE embedding.

# 1 Introduction

Real-world data often exhibit complex structures, making their effective and interpretable visualization a crucial step in exploratory data analysis. Dimensionality reduction methods serve this purpose by projecting high-dimensional data into more interpretable low-dimensional representations while preserving meaningful structures (de Bodt et al., 2025). Among dimensionality reduction methods, principal component analysis (PCA; Hotelling, 1933) and $t$-distributed stochastic neighbor embedding ($t$-SNE; van der Maaten and Hinton, 2008) have emerged as two widely used methods. PCA excels at capturing global structures by projecting data onto a lower-dimensional subspace in directions that maximize variance, providing a broad overview of the dataset's structure. In contrast, $t$-SNE is a neighbor-embedding method with the objective to map data points that are nearby in high-dimensional space close to one another in the low-dimensional embedding. Its focus on neighborhood preservation makes $t$-SNE particularly effective in preserving local structures. When applied individually, both methods suffer from limitations: PCA often overlooks fine local relationships, while $t$-SNE distorts global structures in favor of local neighborhood preservation.

However, in many real-world datasets, e.g., single-cell transcriptomic datasets, both local and global structure is meaningful. Local relationships can reveal microscopic data patterns such as small clusters and cell types. For example, the Tasic et al. dataset features 122 fine clusters that represent cell types, which are clearly separated in the $t$-SNE plot (Figure 1 right). Conversely, the global structure reflects macroscopic data patterns such as separations between broad cell classes (Tasic et al., 2018) or developmental trajectories (Kanton et al., 2019) across an entire cell population. In the Tasic et al. dataset, this level of structure separates individual cells into non-neurons, excitatory, and inhibitory neurons. This global structure is not evident in the $t$-SNE plot, but is prominent in the PCA visualization, which in turn fails to separate the finer clusters (Figure 1 left). Since both local and global structure carry essential and often complementary information, neglecting either structure scale can result in incomplete or biased interpretations, highlighting the importance of preserving both scales simultaneously.

Our proposed method DREAMS combines the interpretability and global structure preservation of PCA with the local sensitivity of $t$-SNE in a simple, yet effective way. On the Tasic et al. data, it maintains PCA's global arrangement of non-neurons, inhibitory, and excitatory neurons, while also revealing the finer cluster structure within each broad group, similar to $t$-SNE (Figure 1 middle). DREAMS integrates global structure preservation into the $t$-SNE objective by a PCA-based regularization term applied throughout the entire optimization process. Varying the regularization strength allows DREAMS to transition from the locally well-structured $t$-SNE embedding to the globally well-structured PCA embedding. Along this spectrum it trades off local and global structure more favorably than its competitors, resulting in improved qualitative and quantitative structure preservation, particularly on datasets with hierarchical organization that exhibit both prominent local and global patterns.

We chose PCA as the default global method in DREAMS due to its simplicity, inherent interpretability (PCA is linear) and fast runtime. That said, DREAMS also offers regularizing with other global embeddings such as multidimensional scaling (MDS).

In summary, we introduce the new method DREAMS making the following contributions:

1. suggest a simple but effective regularization strategy to combine $t$-SNE's local with PCA's global quality;
2. present a spectrum of visualizations with state-of-the-art trade-off between local and global structure;
3. perform a benchmark of ten algorithms on eleven real-world datasets;
4. provide an open-source implementation of DREAMS based on the `openTSNE` library.

Our code is available at `https://github.com/berenslab/dreams-experiments/tree/tmlr`. We used a modified `openTSNE` implementation available at `https://github.com/berenslab/DREAMS/tree/tmlr` and a modified CNE implementation available at `https://github.com/berenslab/DREAMS-CNE/tree/tmlr`.

# 2 Related Work

The most prominent and well-established visualization methods either excel at preserving local structure, e.g., neighbor-embedding methods $t$-SNE (van der Maaten and Hinton, 2008) and UMAP (McInnes et al.,

2018), or global structure, e.g., PCA (Pearson, 1901; Hotelling, 1933) and MDS (Kruskal, 1964), but not both simultaneously. Therefore, many recent efforts have aimed at producing visualizations with faithful local *and* global structure.

Most of them attempt to improve the global structure preservation of neighbor embeddings. One way to make neighbor embeddings more sensitive to the global structure of a dataset is to widen the set of points that are deemed similar beyond a small set of nearest neighbors. For instance, increasing the perplexity parameter in $t$-SNE effectively increases the number of considered nearest neighbors but consequently leads to increased runtime (Kobak and Berens, 2019; Lee et al., 2015; De Bodt et al., 2020). Similarly, several modifications of UMAP, e.g., PaCMAP (Wang et al., 2021) and TriMAP (Amid and Warmuth, 2019), do not only use nearest neighbors, but also consider more distant points in their optimization. In addition to attraction between nearest neighbors, PaCMAP employs weak attraction on mid-near points, while TriMap tries to also preserve the order of similarities in random triplets. Another triplet-based method, ivis (Szubert et al., 2019), strives to balance local and global structure with a parametric encoder and a margin loss on triplets of embedding distances. Changing the similarity of input points more drastically, EmbedOR (Saidi et al., 2025) computes a neighbor embedding from a distance matrix based on the curvature, and thus geometry, of the $k$-nearest neighbor graph.

A different strategy to improve the global structure in neighbor embeddings is to initialize them with a global embedding, which can improve the global structure of the final embedding, despite no further steps for preserving global structure during optimization (Kobak and Linderman, 2021; Wang et al., 2021). UMATO (Jeon et al., 2025) first computes a skeletal layout of only the most densely connected points to capture the global layout and then adds the remaining points in a second optimization phase.

A prominent way to trade off local and global structures within the neighbor-embedding framework is the attraction-repulsion spectrum (Böhm et al., 2022; Damrich et al., 2023). Along this spectrum methods with stronger between-neighbor attraction tend to focus on more global structure. UMAP and $t$-SNE both lie on this spectrum with UMAP having more attraction. The most global method on this attraction-repulsion spectrum is Laplacian Eigenmaps (Belkin and Niyogi, 2003).

Recently, hybrid methods were proposed that combine elements of neighbor embedding methods and global methods during optimization. Several of these are variational autoencoders with a 2D latent space and regularized ELBO maximization. The method scvis (Ding et al., 2018) uses a Gaussian latent prior and adds a variant of the $t$-SNE objective to the ELBO. Its successor VAE-SNE (Graving and Couzin, 2020) employs a more flexible Gaussian mixture prior. Instead of a regularizer promoting local structure preservation, ViVAE (Novak et al., 2023) adds a stochastic MDS regularizer, but denoises the high-dimensional data based on $k$-nearest neighbor relations.

More similar to our approach are non-parametric hybrid methods that directly optimize the embedding positions. Local-to-Global Structures (Miller et al., 2023), a method for generic graph drawing, applies MDS-like stress minimization to shortest path graph distances among pairs of points that are strongly connected, while repelling non-neighboring pairs. Cluster and Embed (Coda et al., 2025) embeds clusters separately, focusing on their local structure. These fixed cluster embeddings then get arranged into a full embedding by rigid transformations that minimize the overall stress like in MDS. The SQuadMDS-hybrid (Lambert et al., 2022) interpolates between the objectives of $t$-SNE and MDS. For MDS it uses the stochastic quartet framework of Lambert et al. (2022). To make both objectives more compatible, SQuadMDS-hybrid normalizes their gradients before blending them together. The hybrid method StarMAP (Watanabe et al., 2025) changes the attractive force in the UMAP objective. Instead of pulling only nearest neighbors together, it also pulls points towards the PCA coordinates of their $k$-means cluster centroid. This combination of a neighbor embedding with PCA is similar to our DREAMS. But instead of UMAP, we use $t$-SNE due to its better local structure preservation. Moreover, we propose a simpler objective that avoids StarMAP's clustering step, leaves the neighbor-embedding objective intact, and simply pulls each embedding point towards its own PCA position. We found that this simpler approach leads to a better local-global trade-off. Additionally, setting the regularization strength in DREAMS to 0 or to 1 allows to recover standard $t$-SNE and PCA, while the StarMAP framework cannot fully recover PCA.

Different from neighbor-embedding-inspired methods, PHATE (Moon et al., 2019) is a diffusion-based method that tries to balance local and global structure preservation. Like neighbor embeddings, it starts with the $k$-nearest neighbor graph of the high-dimensional data. It then integrates this local information into a global graph distance. PHATE uses potential distance, a variant of diffusion distance that focuses more on global structure. To visualize this global distance metric in 2D, PHATE uses MDS. Geometry-regularized autoencoders of Duque et al. (2022) regularize their 2D latent space with a precomputed PHATE embedding. DREAMS also uses a reference embedding for regularization, but is non-parametric, uses PCA or MDS, and employs the $t$-SNE loss instead of the reconstruction loss.

## 3 Background

In dimensionality reduction, we aim to represent a high-dimensional dataset $X = [x_1, \ldots, x_n]^\top \in \mathbb{R}^{n \times m}$ with $n$ observations in an $m$-dimensional space by a lower-dimensional embedding $Y = [y_1, \ldots, y_n]^\top \in \mathbb{R}^{n \times d}$, where $d \ll m$. The objective is to construct $Y$ such that meaningful relationships among the observations are preserved in the lower-dimensional space. In this section, we outline the two dimensionality reduction methods that DREAMS builds upon — PCA and $t$-SNE — and show how they approach this goal from complementary perspectives.

### 3.1 Principal component analysis (PCA)

Principal component analysis (PCA; Pearson, 1901; Hotelling, 1933) is a linear transformation that projects the data onto a new coordinate system aligned with the directions of maximum variance. PCA seeks a linear, orthogonal mapping $W \in \mathbb{R}^{m \times d}$ that projects the data into a lower-dimensional space $Y = XW$, where the directions in $W$ capture the maximal variance across the entire dataset $X$. This ensures the preservation of the macroscopic, global structure since distances along directions with high variance are preserved in the projection (Huang et al., 2022). However, because orthogonal projections can map distant points to similar locations, PCA performs poorly at preserving local structures (Huang et al., 2022; Wang et al., 2023).

### 3.2 $t$-distributed stochastic neighbor embedding ($t$-SNE)

$t$-distributed stochastic neighbor embedding ($t$-SNE; van der Maaten and Hinton, 2008) is a widely used neighbor-embedding method that is especially effective at preserving local similarities (Espadoto et al., 2019). By transforming Euclidean distances between points into pairwise similarity probabilities, $t$-SNE constructs a probability distribution $P$, based on the high-dimensional observations $X$, and a probability distribution $Q$, based on the low-dimensional embeddings $Y$. The distribution $P = \{p_{ij}\}_{i,j=1}^n$ encodes the nearest-neighbor structure in the high-dimensional space via

$$p_{ij} = \frac{p_{j|i} + p_{i|j}}{2n}, \text{ where } p_{j|i} = \frac{\exp\left(-\|x_i - x_j\|^2/(2\sigma_i^2)\right)}{\sum_{k \neq i} \exp\left(-\|x_i - x_k\|^2/(2\sigma_i^2)\right)} \text{ if } i \neq j \text{ and } p_{i|i} = 0.$$

The width $\sigma_i$ of the Gaussian kernels is adaptively chosen for each data point to ensure the same effective neighborhood size, i.e., the number of points $j$ for which $p_{j|i} \gg 0$. Due to the exponential decrease of the Gaussians, most $p_{ij}$ are close to zero, and are treated as exactly zero in most implementations.

The similarity probability distribution of the low-dimensional embedding points $Q = \{q_{ij}\}_{i,j=1}^n$ is based on the Cauchy kernel:

$$q_{ij} = \frac{\left(1 + \|y_i - y_j\|^2\right)^{-1}}{\sum_{k \neq l} \left(1 + \|y_k - y_l\|^2\right)^{-1}} \text{ if } i \neq j \text{ and } q_{ii} = 0.$$

The objective is to arrange the low-dimensional embedding $Y$ such that the low-dimensional similarities $q_{ij}$ match the similarities $p_{ij}$ as measured by the Kullback–Leibler divergence

$$\mathcal{L}_{t\text{-SNE}}(Y) = \text{KL}(P \parallel Q) = \sum_{i,j} p_{ij} \log \frac{p_{ij}}{q_{ij}}, \tag{1}$$

which is minimized via gradient descent with respect to the embedding positions $Y$.

The similarity probabilities act as kernels centered around the data and embedding points, assigning high similarity values only to close neighbors, while distant points have little impact on the loss function. This makes neighbors in the high-dimensional space remain neighbors in the low-dimensional embedding space, ensuring the preservation of local structure. In contrast, due to the weak influence of distant points, global structure can be misrepresented (Huang et al., 2022; Wang et al., 2023).

## 4 Methods

### 4.1 Regularizing with precomputed global embedding

In DREAMS, we first precompute the PCA positions $\widetilde{Y} \in \mathbb{R}^{n \times 2}$ as the reference embedding for the global structure of the data. To combine the local structure preservation of $t$-SNE with the global structure preservation of PCA, we augment the $t$-SNE loss function by a regularization term that penalizes embedding points $Y$ deviating from their PCA positions $\widetilde{Y}$, yielding the loss

$$\mathcal{L}(Y) = (1 - \lambda)\mathcal{L}_{t\text{-SNE}}(Y) + \frac{\lambda}{n}\|Y - \alpha\widetilde{Y}\|_F^2. \tag{2}$$

Since a PCA embedding scales linearly with the original data scale, while a $t$-SNE embedding does not, we rescale $\widetilde{Y}$ to match the scale of $Y$ during each gradient-descent iteration by computing a scalar $\alpha$

$$\alpha = \|Y\|_F / \|\widetilde{Y}\|_F. \tag{3}$$

This scaling encourages the reference embedding to match the current scale of the $t$-SNE embedding during optimization, making the global and local objectives more compatible. Note that $\alpha$ was treated as a constant in each gradient descent iteration. We discuss alternative scaling options in Appendix B.

The first term of Equation 2 enforces the preservation of local structure in the embedding by minimizing the $t$-SNE loss (Equation 1), while the regularization term ensures the preservation of global structure by encouraging the embedding to resemble the (scaled) PCA embedding $\widetilde{Y}$. This setup allows for local adjustments by the $t$-SNE loss, while the quadratic penalty prevents large deviations that would distort the global layout. The regularization strength $\lambda$ controls the impact of the PCA embedding on the final embedding, thereby enabling a trade-off between local and global structure preservation. For $\lambda = 0$, DREAMS produces a standard $t$-SNE embedding since the loss term (Equation 2) reduces to the $t$-SNE objective. In the other limiting case of regularization strength $\lambda = 1$, the objective becomes the regularizer without the $t$-SNE loss whose optimum is a (scaled) PCA embedding. For intermediate $\lambda \in (0, 1)$ the objective is a weighted mean between the $t$-SNE objective and the regularizer. Empirically, we found $\lambda = 0.15$ to be a suitable value for combining the individual strengths of global structure preservation of PCA with the local structure preservation of $t$-SNE, without substantially compromising either aspect.

This framework is not limited to using the PCA embedding as the global reference embedding $\widetilde{Y}$. Depending on the characteristics of the dataset, other methods with good global structure preservation, such as MDS, can also be used as an effective choice for $\widetilde{Y}$. We will refer to the version of DREAMS that uses the SQuadMDS embedding for regularization as DREAMS-MDS. By default, we use PCA instead of MDS because PCA is faster and is inherently interpretable due to its linearity.

Our implementation is based on the `openTSNE` library (Poličar et al., 2024), where we added the regularization term to the gradient-descent optimization. We used default `openTSNE` hyperparameters and initialized the embedding with the regularization embedding $\widetilde{Y}$, which was rescaled so that its first dimension had a standard deviation of $10^{-4}$ (as is default in `openTSNE`).

The `openTSNE` library implements Barnes–Hut $t$-SNE (Yang et al., 2013; Van Der Maaten, 2014) and FI$t$-SNE (Linderman et al., 2019) approximations having runtime complexity $\mathcal{O}(n \log n)$ and $\mathcal{O}(n)$ respectively.

### 4.2 Linear decoding regularization

An alternative approach to using a precomputed PCA embedding is using a linear decoder. Equivalent to variance maximization, PCA can also be obtained by minimizing the reconstruction error

$$\widetilde{W} = \underset{W \in \mathbb{R}^{m \times d}}{\arg\min} \|X - XWW^\top\|_F^2 \text{ subject to } W^\top W = I_d. \tag{4}$$

Since the PCA embedding is given by $\widetilde{Y} = X\widetilde{W}$, the DREAMS loss with linear decoding regularizer becomes

$$\mathcal{L}(Y, D) = (1 - \lambda)\mathcal{L}_{t\text{-SNE}} + \lambda \left\| X - (YD^\top + b) \right\|_F^2,$$

with $D \in \mathbb{R}^{m \times d}$ being a trainable linear decoder and $b \in \mathbb{R}^m$ a trainable bias term that is added row-wise and allows to handle uncentered embeddings. In this setup, by minimizing the reconstruction error, the decoder is responsible for the global structure preservation by pushing the embedding towards the PCA structure that has the minimal reconstruction error. If $D$ were constrained to have orthogonal columns ($D^\top D = I_d$), the optimum of the regularization term would be $D = \widetilde{W}$ (Plaut, 2018; Nazari et al., 2023). Although we do not explicitly enforce orthogonality, we observed that the learned linear decoder $D$ naturally tends to have approximately orthogonal columns.

We based our implementation on InfoNC-$t$-SNE (Damrich et al., 2023), a GPU-based contrastive learning approximation of $t$-SNE with the InfoNCE loss, implemented in PyTorch as CNE (contrastive neighbor embedding) package. This allowed us to add the decoding regularizer based on a linear PyTorch layer. We increased the number of negative samples to 500 to improve the local structure preservation and approximate $t$-SNE more closely. For the same reason, we ran the optimization for 750 epochs. The remaining hyperparameters were kept at default values. For the regularization term, we used a linear layer mapping the low-dimensional embedding to the original feature space. The weights were initialized using the first two principal components of the data (the bias term was initialized with zero). We will refer to this version as DREAMS-CNE-Decoder.

For comparison, we also implemented a version of DREAMS using the CNE backend with precomputed PCA regularization as in Section 4.1. We will refer to this version as DREAMS-CNE. Here the gradient with respect to $Y$ was computed using autodifferentiation, including the $\|Y\|_F$ contribution to $\alpha$.

## 5 Experimental setup

### 5.1 Datasets and performance metrics

To validate our method experimentally, we used eleven real-world datasets, all but one containing both prominent local and global structures (Table 1). We measured the embedding quality using two established metrics quantifying local and global structure preservation (Kobak and Berens, 2019):

KNN The $k$-nearest neighbor recall (KNN) is the fraction of $k$-nearest neighbors in the high-dimensional data that are preserved as $k$-nearest neighbors in the low-dimensional embedding. We used $k = 10$ throughout all experiments (for results with different choices of $k$, see Figure S5). The final metric is given as the average across all $n$ data points. KNN quantifies the preservation of local structure in the embedding.

CPD The correlation of pairwise distances (CPD) is the Spearman correlation between the pairwise distances in the high-dimensional space and in the embedding. We computed pairwise distances among $1\,000$ randomly chosen data points. CPD quantifies the preservation of the global structure in the embedding.

We also evaluated an aggregated local-global score $s$. For each dataset, we normalized both metrics based on the minimum and the maximum values observed across all methods. Let $\text{KNN}_{\min}$ and $\text{CPD}_{\min}$ denote the lowest (worst) KNN and CPD scores across all methods for a given dataset, and $\text{KNN}_{\max}$ and $\text{CPD}_{\max}$

Table 1: Datasets used in our experiments with a description of their local and global structure. For MNIST and Fashion MNIST, we only used the training set.

| Name | Description | Global structure | Local str. | $n$ |
|------|-------------|------------------|-----------|-----|
| Tasic et al. | scRNA-seq of mouse cortex | major cell classes | cell types | 23 822 |
| Macosko et al. | scRNA-seq of mouse retina | major cell classes | cell types | 44 808 |
| Kanton et al. | scRNA-seq of human brain | developmental trajectory | cell types | 20 272 |
| Wagner et al. | scRNA-seq of zebrafish embryos | developmental trajectory | cell types | 63 530 |
| Packer et al. | scRNA-seq of C. elegans | developmental trajectory | cell types | 86 024 |
| 1000 Genomes | human whole-genome sequencing | continental ancestry | populations | 3 450 |
| Mammoth | 3D Mammoth skeleton | overall body shape | bone structure | 50 000 |
| Satellite | satellite image crops | soil color | 6 soil types | 6 435 |
| FMNIST | images of fashion items | shoes/bags/garments | 10 classes | 60 000 |
| MNIST | hand-written digits | none | digits 0–9 | 60 000 |
| CIFAR10 | vehicle and animal images | vehicles vs animals | 10 classes | 60 000 |

denote the highest (best) scores. Given an embedding with specific KNN and CPD values, we define the aggregated local-global score as

$$s = \frac{1}{2} \left( \frac{\text{KNN} - \text{KNN}_{\min}}{\text{KNN}_{\max} - \text{KNN}_{\min}} + \frac{\text{CPD} - \text{CPD}_{\min}}{\text{CPD}_{\max} - \text{CPD}_{\min}} \right). \tag{5}$$

This score ranges from 0 (embedding has the worst local and the worst global scores) to 1 (embedding has the best local and the best global scores) and allows direct comparison of embedding methods regarding their combined local and global structure preservation. Note that it only produces a relative score as it depends on the considered competitors. For an intuitive understanding of the local-global score, see Figure S4.

These metrics rely solely on the high-dimensional data $X$ and its corresponding embedding $Y$ and do not make use of any metadata listed in Table 1. In contrast, metrics like classification accuracy, or class separation measured via the Silhouette score, or agreement between clusters and classes measured via the adjusted Rand index, all require class labels.

## 5.2 Comparison methods

We validated DREAMS against several baseline and hybrid methods (always using the default hyperparameter settings):

- $t$-SNE (van der Maaten and Hinton, 2008) using the `openTSNE` implementation (Poličar et al., 2024) and UMAP (McInnes et al., 2018) as baselines for local structure preservation;

- PCA and MDS using the SQuadMDS implementation (Lambert et al., 2022) as baselines for global structure preservation;

- TriMap (Amid and Warmuth, 2019), PacMAP (Wang et al., 2023), and PHATE (Moon et al., 2019) as methods that strive to preserve both local and global structure;

- and hybrid approaches SQuadMDS-hybrid (Lambert et al., 2022) and StarMAP (Watanabe et al., 2025), which, like DREAMS, mix the local structure preservation of neighbor embeddings with global methods.

We always report means and uncertainties (often barely perceptible) across four random seeds. All experiments were conducted on a single Intel Xeon Gold 6226R CPU 2.90 GHz (16 cores, 32 threads), an NVIDIA RTX A6000 GPU (48 GB VRAM, CUDA 12.7), and 377 GB system RAM, running on a Linux environment. Only experiments including DREAMS-CNE or DREAMS-CNE-Decoder utilized the GPU. Without parallelization, the runtime of all experiments was about a week. For a runtime analysis of DREAMS and its competitors, see Figure S2.

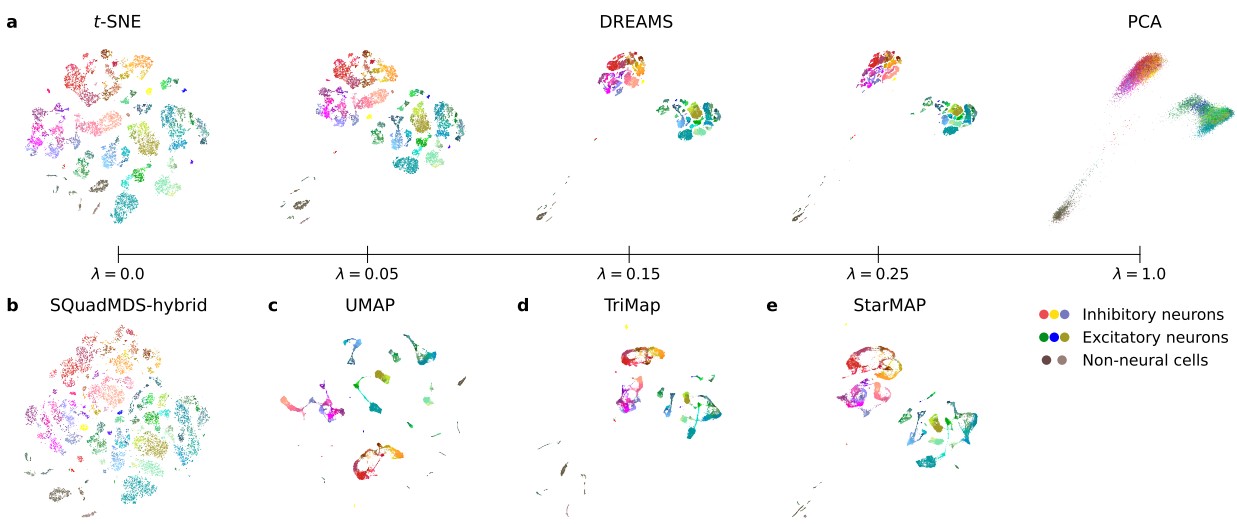

Figure 2: Embeddings of the Tasic et al. dataset. **a:** Spectrum of DREAMS embeddings for different values of regularization strength $\lambda$. **b–e:** Embeddings obtained by some of the competing methods. For all embeddings, see Figure S6.

## 6 Results

### 6.1 DREAMS successfully combines the strengths of $t$-SNE and PCA

We first illustrate DREAMS using the Tasic et al. dataset. When adjusting the regularization strength $\lambda$, DREAMS generates a continuum of embeddings between the two extremes of the locally focused $t$-SNE embedding and the globally focused PCA embedding (Figure 2a). For small $\lambda$ values, the resulting embedding resembled the $t$-SNE embedding, effectively capturing the local data structure and highlighting fine-grained clusters and cell types. For larger $\lambda$ values, the embedding progressively shifted towards the PCA embedding, emphasizing higher-level groupings, such as broad cell classes (inhibitory/excitatory neurons and non-neuronal cells). At intermediate regularization strengths, DREAMS integrated both local and global structures without visibly compromising either aspect of the data. Competing methods often missed much of the global structure (Figure 2b,c) or showed less local structure (Figure 2c–e).

The quantitative metrics corroborated the ability of DREAMS to maintain both local and global structure (Figure 3). On all datasets but Mammoth, $t$-SNE achieved the highest KNN value, providing the best locally structured embedding. In contrast, PCA and MDS maintained global structure best, as reflected in their CPD values being the highest. For nearly all regularization parameters $\lambda \in [0, 1]$ DREAMS yielded embeddings with better local or global structure preservation than its competitors. On most datasets, DREAMS with its default regularization strength ($\lambda = 0.15$) simultaneously achieved KNN close to $t$-SNE's and CPD close to PCA's. Across all datasets, DREAMS preserved the local structure much better than StarMAP, which also combines neighbor embeddings with PCA. This is likely because DREAMS relies on $t$-SNE, which preserves local structure better than UMAP, which is used in StarMAP.

UMAP generally performed poorly in our metrics, often preserving not only the local but also the global structure worse than $t$-SNE, as measured by the CPD metric. This may be due to different default initializations of `openTSNE` and UMAP (PCA and Laplacian Eigenmaps, respectively). That said, note that UMAP is known to perform well in clustering-based metrics (Espadoto et al., 2019; Xiang et al., 2021; Lause et al., 2024), which we do not use for evaluation in this work.

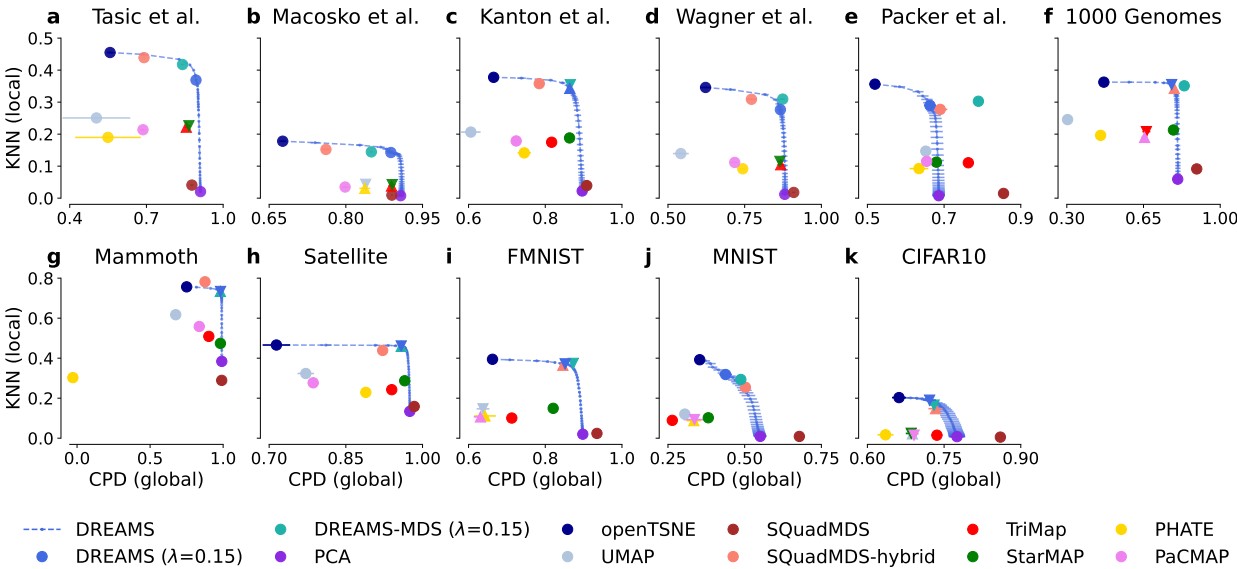

Figure 3: Quantitative evaluation of local and global structure preservation of different methods across multiple datasets (Table 1). Spearman correlation of pairwise distances (CPD, global metric) is plotted against $k$NN recall (KNN, local metric). For improved visual clarity different markers were used.

TriMap performed similarly well to StarMap (note that both use PCA as initialization), but still fell short compared to DREAMS in terms of local structure preservation. PHATE and PacMAP yielded worse CPD and worse KNN scores compared to DREAMS on all datasets.

On the MNIST dataset DREAMS was not able to maintain high global and local structure at the same time, likely because this dataset does not have a prominent global structure in the first place (CPD values were comparatively low for all methods including PCA and MDS). Similarly, on the pixel-valued CIFAR10 dataset, DREAMS incurred a clear trade-off between local and global structure. On this challenging dataset all methods severely struggled (Figure S16). Nevertheless, even in these two cases, DREAMS was on par or outperformed most other methods (Figure 3j,k).

Using our aggregated local-global score, we could directly compare DREAMS to other methods along a single dimension (Table 2). The highest local-global score on every dataset was achieved by either DREAMS or DREAMS-MDS, depending on whether PCA or MDS was capturing global structure more effectively in terms of the CPD metric. Following DREAMS, SQuadMDS-hybrid was consistently the next best method. It achieved scores comparable to DREAMS on three datasets, but on all remaining datasets, its score was lower by at least 0.06 than the best performing DREAMS variant. On the datasets where DREAMS achieved a good balance between local and global performance (all but MNIST and CIFAR10), all other methods had scores lower by at least 0.09 than the best performing DREAMS variant.

## 6.2 DREAMS provides the best local-global spectrum

Next, we compared the local-global embedding spectrum of DREAMS to the local-global spectra of existing methods and variants of DREAMS.

In SQuadMDS-hybrid, the spectrum of embeddings is obtained by specifying the learning rates of $t$-SNE and MDS. For StarMAP, varying the regularization strength allows trading off local and global structure. Furthermore, increasing the exaggeration parameter in `openTSNE` can emphasize global structure preservation and also yields a local-global continuum of embeddings (Böhm et al., 2022).

Table 2: Aggregated local-global score. For each dataset, the methods within 0.05 of the highest score are highlighted in bold. The order of the datasets is as in Table 1.

|              | Tas  | Mac  | Kan  | Wag  | Pac  | 1kG  | Mam  | Sat  | FMN  | MNI  | C10  |
|--------------|------|------|------|------|------|------|------|------|------|------|------|
| DREAMS       | **0.88** | **0.85** | **0.88** | 0.84 | 0.62 | **0.89** | **0.95** | **0.95** | 0.83 | **0.61** | **0.66** |
| DREAMS-MDS   | **0.87** | 0.78 | **0.90** | **0.90** | **0.83** | **0.93** | **0.95** | **0.94** | **0.87** | **0.64** | **0.62** |
| SQuadMDS-hybrid | 0.71 | 0.61 | 0.77 | 0.76 | 0.64 | **0.88** | **0.94** | 0.84 | 0.81 | **0.61** | 0.58 |
| StarMAP      | 0.68 | 0.57 | 0.66 | 0.59 | 0.39 | 0.66 | 0.68 | 0.70 | 0.48 | 0.26 | 0.16 |
| PHATE        | 0.25 | 0.42 | 0.40 | 0.39 | 0.29 | 0.35 | 0.01 | 0.47 | 0.14 | 0.19 | 0.03 |
| PaCMAP       | 0.45 | 0.34 | 0.42 | 0.39 | 0.36 | 0.51 | 0.70 | 0.35 | 0.12 | 0.20 | 0.15 |
| TriMap       | 0.66 | 0.55 | 0.56 | 0.58 | 0.51 | 0.55 | 0.68 | 0.58 | 0.24 | 0.10 | 0.25 |
| UMAP         | 0.26 | 0.46 | 0.26 | 0.19 | 0.40 | 0.31 | 0.68 | 0.39 | 0.18 | 0.19 | 0.16 |
| *t*-SNE      | 0.57 | 0.50 | 0.60 | 0.61 | 0.50 | 0.64 | 0.86 | 0.50 | 0.55 | **0.61** | 0.56 |
| PCA          | 0.50 | 0.50 | 0.48 | 0.46 | 0.25 | 0.43 | 0.60 | 0.48 | 0.44 | 0.35 | 0.32 |
| MDS          | 0.48 | 0.47 | 0.52 | 0.51 | 0.51 | 0.55 | 0.50 | 0.54 | 0.51 | 0.50 | 0.50 |

We observed that these spectra offered noticeably worse trade-offs than DREAMS, as illustrated on the Tasic et al. dataset in Figure 4a, where PCA produces the best global layout. Similarly, DREAMS-MDS achieves a better trade-off than SQuadMDS-hybrid, as demonstrated on the Packer et al. dataset (Figure 4b), where MDS produces the best global layout.

Moreover, while DREAMS includes standard $t$-SNE and PCA/MDS as its corner cases for $\lambda = 0$ and $\lambda = 1$ (marked with stars in Figure 4), neither SQuadMDS-hybrid nor StarMAP could reach MDS and PCA, respectively, in their most global setting. Additionally, SQuadMDS-hybrid underperformed compared to $t$-SNE in its most local configuration (Figure 4a,b). This outcome is expected for StarMAP, as its objective continues to blend aspects of UMAP and PCA even at its highest regularization strength. For SQuadMDS-hybrid, the reason is likely the normalization of the $t$-SNE and MDS gradients before combining them.

Switching the neighbor-embedding backend of DREAMS from `openTSNE` to contrastive neighbor embeddings (CNE; Damrich et al., 2023) with the InfoNCE loss decreased the KNN score, proportionally to the difference in KNN score between $t$-SNE and its InfoNCE version InfoNC-$t$-SNE (Figure 4c). Using the CNE backend, we compared regularization using a precomputed PCA embedding (DREAMS-CNE) with linear decoding regularization (DREAMS-CNE-Decoder) and observed only marginal improvements with the decoder approach (Figure S3). The MNIST dataset was the only one where the decoder approach performed much better in terms of our metrics, but this did not translate into visible improvements of the embedding structure. Although the decoder could reconstruct the PCA embedding and enhance global structure, it introduces additional randomness and computational complexity. Moreover, with a precomputed global embedding, we are more flexible to adjust the global layout, e.g., by switching from PCA to MDS. For these reasons and especially due to the better local structure of `openTSNE` than InfoNC-$t$-SNE, we prefer the simple regularization using a precomputed embedding as in the `openTSNE`-based DREAMS variants.

## 7    Discussion

In this work, we introduced DREAMS, a dimensionality reduction method that adds a regularization term to the $t$-SNE objective. This regularization term penalizes embedding points that deviate from their PCA positions and thereby encourages global structure preservation throughout the optimization process. Our approach addresses a critical shortcoming in conventional dimensionality reduction techniques, which often prioritize one structural scale over the other. Through this simple yet effective regularization term and a tunable hyperparameter $\lambda$, DREAMS allows for a continuous spectrum of embeddings that transition smoothly from the locally faithful structure of $t$-SNE to the globally coherent and interpretable structure of PCA. Furthermore, with its default regularization strength ($\lambda = 0.15$), DREAMS provides an embedding that successfully balances local and global structure preservation, outperforming competing methods.

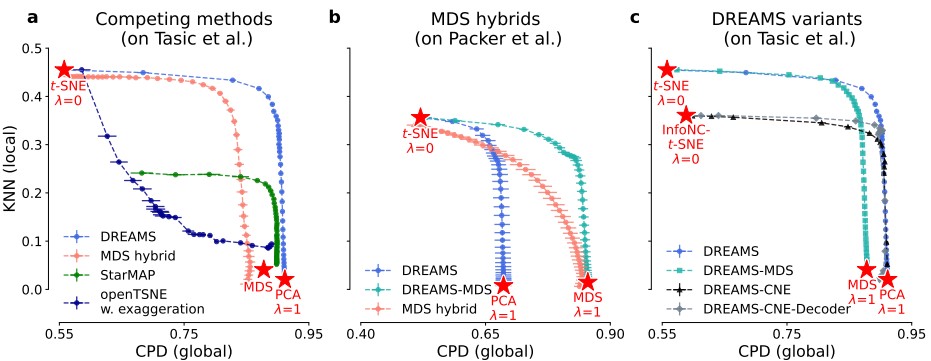

Figure 4: Trade-offs between local and global structure preservation in different methods. Spearman correlation of pairwise distances is plotted against $k$NN recall. **a:** Performance of DREAMS compared with other local-global spectra. **b:** Comparison of DREAMS-MDS and SQuadMDS-hybrid. **c:** Performance of different DREAMS variants. Panels **a** and **c** show results on the Tasic et al. dataset, while panel **b** is based on the Packer et al. dataset.

Across eleven real-world datasets, DREAMS or DREAMS-MDS with default $\lambda = 0.15$ consistently achieved the best performance in terms of the local-global score. This highlights its robustness and wide applicability, especially to hierarchical datasets with multiple inherent structure scales. While multiscale structure is common in real-world datasets, DREAMS is less useful in datasets without it, such as the MNIST dataset, as there is less global structure to preserve. In particular, we do not have theoretical performance guarantees ensuring that DREAMS will balance local and global structure well. The additional time to compute the regularizer and its gradient led to DREAMS having slower runtime than most competitors (Figure S2). Moreover, DREAMS introduces an additional hyperparameter, the regularization strength $\lambda$, whose optimal value can depend on the specific dataset and the intended use. Consequently, different datasets may require tuning $\lambda$ to achieve the optimal balance between local and global structure preservation. That said, we found that $\lambda = 0.15$ worked well across all datasets.

Because the regularization enforces alignment with a predefined structure, any biases or limitations in the PCA embedding would propagate into the final embedding. As demonstrated in cases where the MDS embedding captures global structure more effectively than PCA, it can sometimes be beneficial to regularize towards the MDS embedding, as in DREAMS-MDS. Thanks to its flexible and simple design, DREAMS accommodates such substitutions, making it adaptable to the specific characteristics of different datasets.

Our alternative implementation DREAMS-CNE-Decoder performed worse than the default DREAMS due to the lower local quality of sampling-based InfoNC-$t$-SNE. Therefore, we prefer our `openTSNE`-based implementation, even though the PyTorch implementation may be preferable in some use cases.

In conclusion, DREAMS successfully combines the local structure preservation strength of $t$-SNE with the global structure of PCA. This makes visualizations of high-dimensional data more faithful and interpretable, particularly for datasets with hierarchical structure.

## Acknowledgments

We thank Pierre Lambert and Cyril de Bodt for their help with SQuadMDS, Koshi Watanabe for making the StarMAP code available, and Alex Diaz-Papkovich for providing the preprocessed 1000 Genomes data.

This work was funded by the Gemeinnützige Hertie-Stiftung. DK is a member of the Germany's Excellence cluster 2064 "Machine Learning — New Perspectives for Science" (EXC 390727645). The National Institutes of Health (UM1MH130981) funded initial phases of this work. The content is solely the responsibility of the authors and does not necessarily represent the official views of the National Institutes of Health.

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

## Appendix

## A   Datasets

All scRNA-seq datasets (except Packer et al.) were preprocessed as in Böhm et al. (2022) and Kobak and Berens (2019). After selecting the 1000 (3000 for Macosko et al.) most variable genes, we normalized the library sizes to the median library size in the dataset, log-transformed the normalized values with $\log_2(x+1)$, and finally reduced the dimensionality to 50 via PCA. The Packer et al. dataset was already preprocessed to 100 principal components of which we used the first 50. The original data was downloaded following links in the original publications.

The raw 1000 Genomes Project data (The 1000 Genomes Project Consortium, 2015) is available at `https://ftp.1000genomes.ebi.ac.uk`. This dataset contains $3\,450$ human genotypes. We got the preprocessed data from Diaz-Papkovich et al. (2019; 2023); this is in an integer-valued data matrix with $53\,999$ features, containing values 0, 1, and 2, representing the number of alleles differing from a reference genome. We used PCA to reduce the number of features to 50.

We used $50\,000$ unprocessed random samples of the Mammoth dataset (Smithsonian Institution, 2020; Noichl, 2025) and likewise used the unprocessed Landsat Satellite (Srinivasan, 1993) dataset. For the Mammoth dataset, cluster labels (solely used for coloring the visualizations) were obtained by applying Agglomerative Clustering with 11 classes to the original data.

For the MNIST (Lecun et al., 1998), Fashion MNIST (Xiao et al., 2017), and CIFAR10 (Krizhevsky et al., 2009) datasets (downloaded using the torchvision API), we used the first 50 principal components. For Fashion MNIST, we additionally rescaled the original data to the interval $[0,1]$ before computing the first 50 principal components.

## B   Role of global embedding scaling in the DREAMS loss

In the regularization term

$$\mathcal{R}(Y) = \frac{1}{n}\|Y - \alpha\widetilde{Y}\|_F^2, \tag{6}$$

the scaling parameter $\alpha$ is used to match the scale of the global reference embedding $\widetilde{Y}$ with that of $Y$. We considered two scaling strategies:

$$\alpha_n = \frac{\|Y\|_F}{\|\widetilde{Y}\|_F}, \quad \alpha_p = \frac{\langle Y, \widetilde{Y}\rangle_F}{\|\widetilde{Y}\|_F^2}. \tag{7}$$

By the Cauchy–Schwarz inequality, $\alpha_n \geq \alpha_p$, with equality if and only if $Y$ is a positive rescaling of $\widetilde{Y}$. The first option, $\alpha_n$, rescales $\widetilde{Y}$ to match the scale of $Y$, preventing the regularization term from imposing an artificial scale onto the scale-sensitive $t$-SNE embedding. The second option, $\alpha_p$, is the $\alpha$ that minimizes the regularization term $\mathcal{R}(Y)$ for a given $Y$ and $\widetilde{Y}$ and is inspired by Procrustes analysis (Goodall, 1991; Gower, 1975).

Experimentally, we found that $\alpha_p$ performed slightly worse than $\alpha_n$ on the Tasic et al. dataset, leading to reduced local structure preservation as measured by a smaller KNN value (Table S1). While the regularization term was, by construction, smaller with $\alpha_p$, the KL divergence was slightly higher and the scale of the final embedding was further from $t$-SNE's than with $\alpha_n$ (Figure S1). Consequently, we chose to use the scaling with $\alpha_n$ and conducted all subsequent experiments with this scaling method. Importantly, both scaling strategies worked much better than not using scaling at all and fixing $\alpha = 1$ (Figure S1, Table S1).

Treating $\alpha$ as a constant during optimization yields the gradient

$$\nabla_Y \mathcal{R}(Y) = \frac{2}{n}\left(Y - \alpha\widetilde{Y}\right), \tag{8}$$

Table S1: Comparison of DREAMS performance across various alignment approaches for matching $\widetilde{Y}$ and $Y$ on the Tasic et al. dataset. We compare default DREAMS with DREAMS using a lower scaling factor $\alpha_p$, additionally translationally aligned the global reference $\tilde{Y}$ to $Y$ (treating the shift as constant), and full Procrustes analysis, which also contains a rotational alignment. The smaller scaling with $\alpha_p$ was detrimental, while translational and rotational alignment had no effect as the $t$-SNE loss is invariant to them.

| Metric | DREAMS ($\alpha_n$) | $\alpha_p$ | $\alpha_p$ + translation | full Procrustes | no scaling |
|--------|---------------------|------------|--------------------------|-----------------|------------|
| KNN | **0.37** | 0.35 | 0.35 | 0.35 | 0.27 |
| CPD | **0.89** | **0.89** | **0.89** | **0.89** | 0.53 |

which pulls $Y$ towards the scaled global reference embedding $\alpha\widetilde{Y}$. In contrast, using $\alpha_n$ scaling and allowing the gradients to propagate through $\alpha$, yields the gradient

$$\nabla_Y \mathcal{R}(Y) = \frac{2}{n}\left(Y - \alpha\tilde{Y}\right) - \frac{2}{n}\left(\frac{\langle Y, \tilde{Y}\rangle}{\|Y\|_F \|\tilde{Y}\|_F} - 1\right)Y, \tag{9}$$

with an additional second term that corresponds to a shrinkage of $Y$ towards the origin, since $\frac{\langle Y,\tilde{Y}\rangle}{\|Y\|_F\|\tilde{Y}\|_F} \leq 1$. As mentioned above, since the $t$-SNE loss is scale-sensitive, we do not want the regularizer to have a bearing on the scale of $Y$. Nevertheless, in the CNE variant of DREAMS, we propagated the gradient through $\alpha$ and still obtained a good trade-off between the InfoNC-$t$-SNE and PCA performance (Figure 4). We also verified that treating $\alpha$ as a constant during optimization in the CNE version did not lead to any noticeable difference in performance.

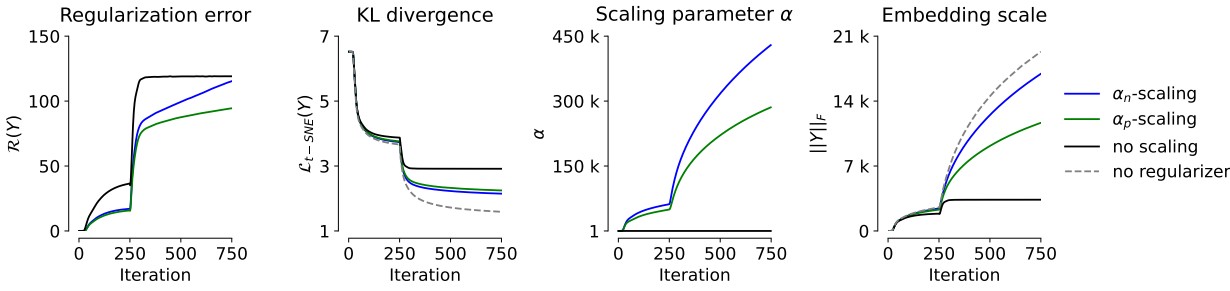

Figure S1: Regularization error, KL divergence, scaling parameter $\alpha$ and embedding scale of DREAMS during optimization using different or no scaling methods. Results are reported as the mean over four random seeds on the Tasic et al. dataset.

## C  Runtime analysis

DREAMS' adds a small overhead for computing the gradients of the regularizer compared to `openTSNE`. As a result, it runs bit slower than `openTSNE` and similarly fast as SQuadMDS-hybrid. The other methods were faster than DREAMS (Figure S2).

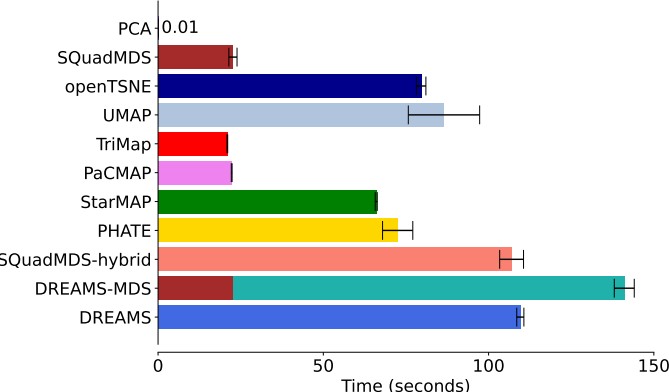

Figure S2: Runtime comparison. The bars indicate the mean runtime and standard deviation across four random seeds on the Tasic et al. dataset. For DREAMS and DREAMS-MDS, the reported times also include the runtime of the respective global reference embeddings (PCA and MDS).

## D    DREAMS-CNE and DREAMS-CNE-Decoder

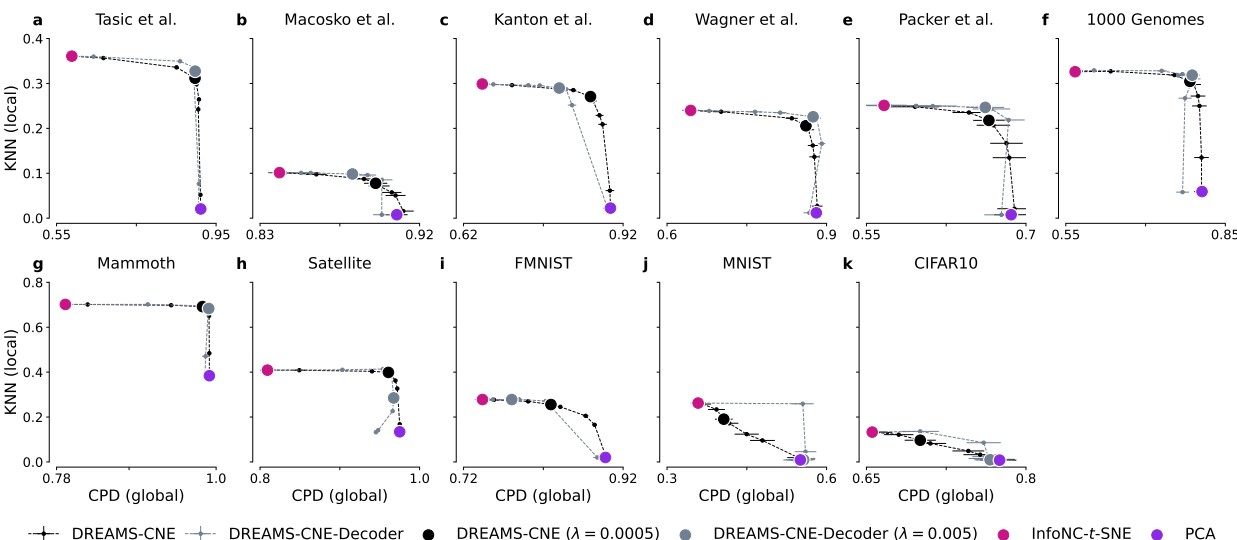

Figure S3: Spearman correlation of pairwise distances (CPD, global metric) is plotted against $k$NN recall (local metric). The figure shows the spectrum of DREAMS-CNE, using a regularizer with precomputed PCA embedding, and DREAMS-CNE-Decoder, using a regularizer with a linear decoder, across all eleven datasets. The bigger scatter points display DREAMS-CNE and DREAMS-CNE-Decoder with their respective default regularization strengths (which achieved the highest average local-global score across all data sets) and local (InfoNC-$t$-SNE) and global (PCA) reference methods. Here, InfoNC-$t$-SNE is used as the $t$-SNE backbone and corresponds to the regularization strength of $\lambda = 0$ while PCA corresponds to the maximal regularization strength of $\lambda = 1$. In panel j (MNIST) the scatter points of PCA and DREAMS-CNE-Decoder lie exactly on top of each other.

# E    Intuition of local-global score

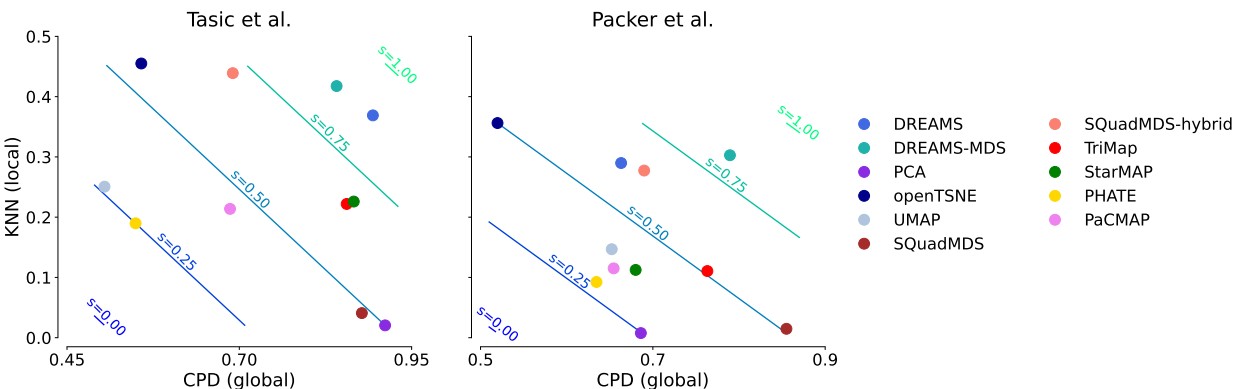

Figure S4: Spearman correlation of pairwise distances (CPD, global metric) is plotted against kNN recall (KNN, local metric) for different methods with marked local-global score spectrum. The figure shows the structure preservation results on the Tasic et al. and Packer et al. dataset.

# F    Sensitivity of $k$NN recall to choice of $k$

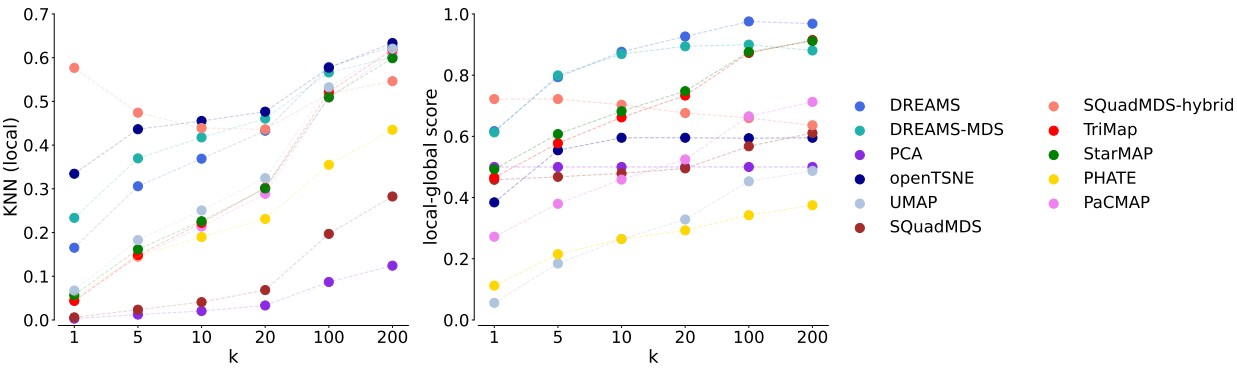

Figure S5: Local structure preservation metric and combined local-global score for different numbers of neighbors $k$ in the $k$NN recall. Displayed are the results on the Tasic et al. dataset. Except for SquadMDS-hybrid, the ranking of methods in terms of $k$NN recall is insensitive to the exact value of $k$.

## F.1    Visualizations of all datasets

See next pages.

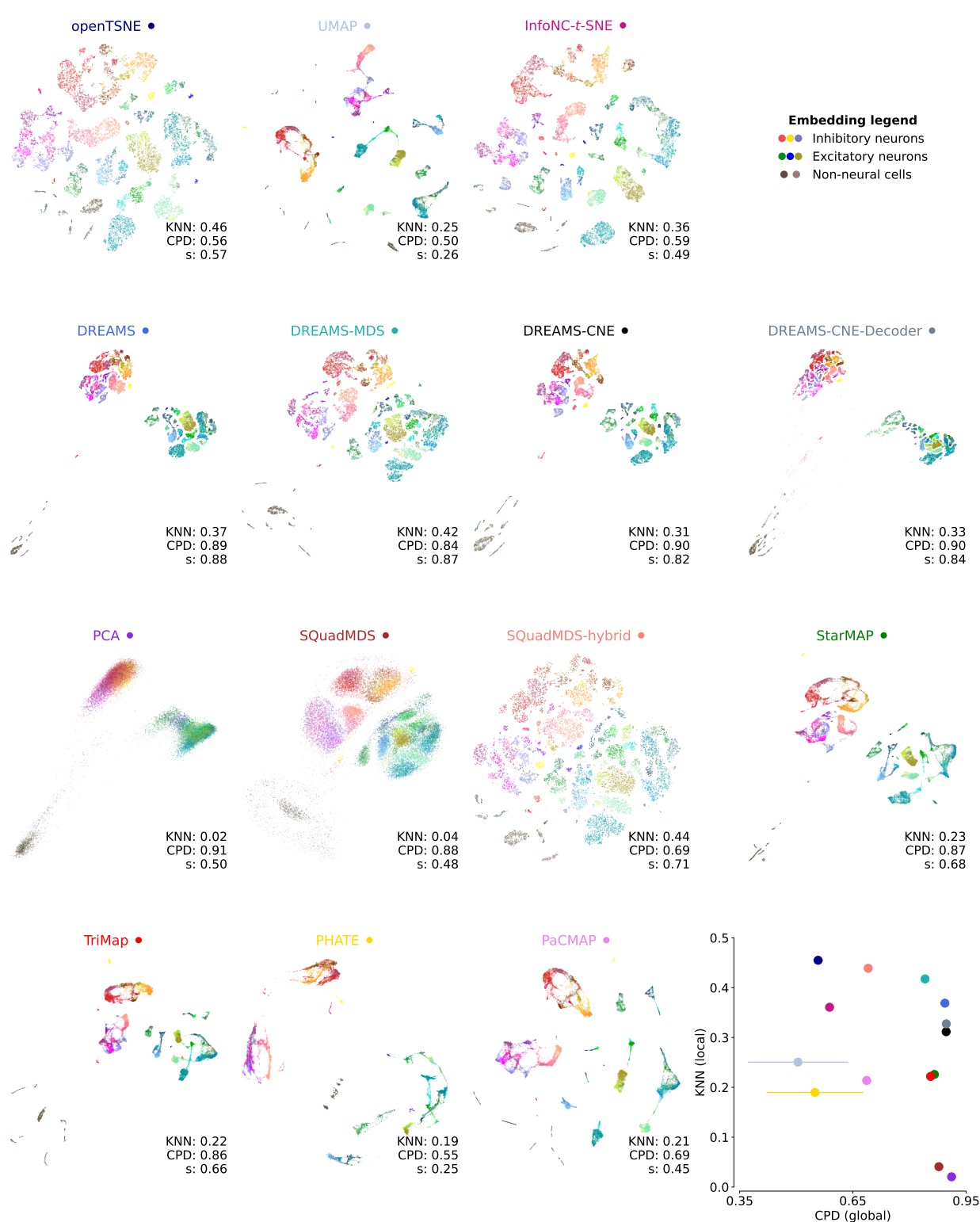

Figure S6: Visualizations of the Tasic et al. dataset with all considered methods.

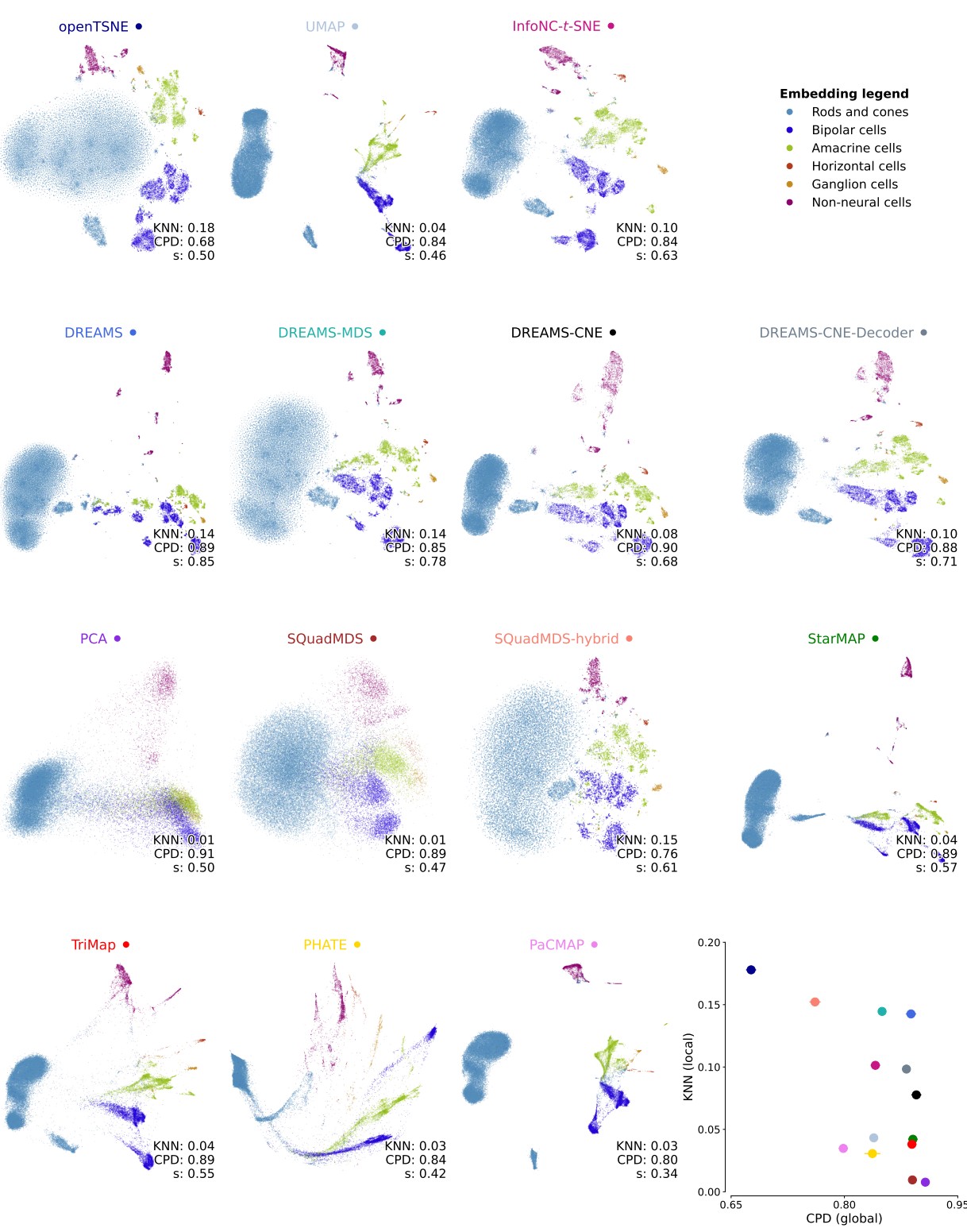

Figure S7: Visualizations of the Macosko et al. dataset with all considered methods.

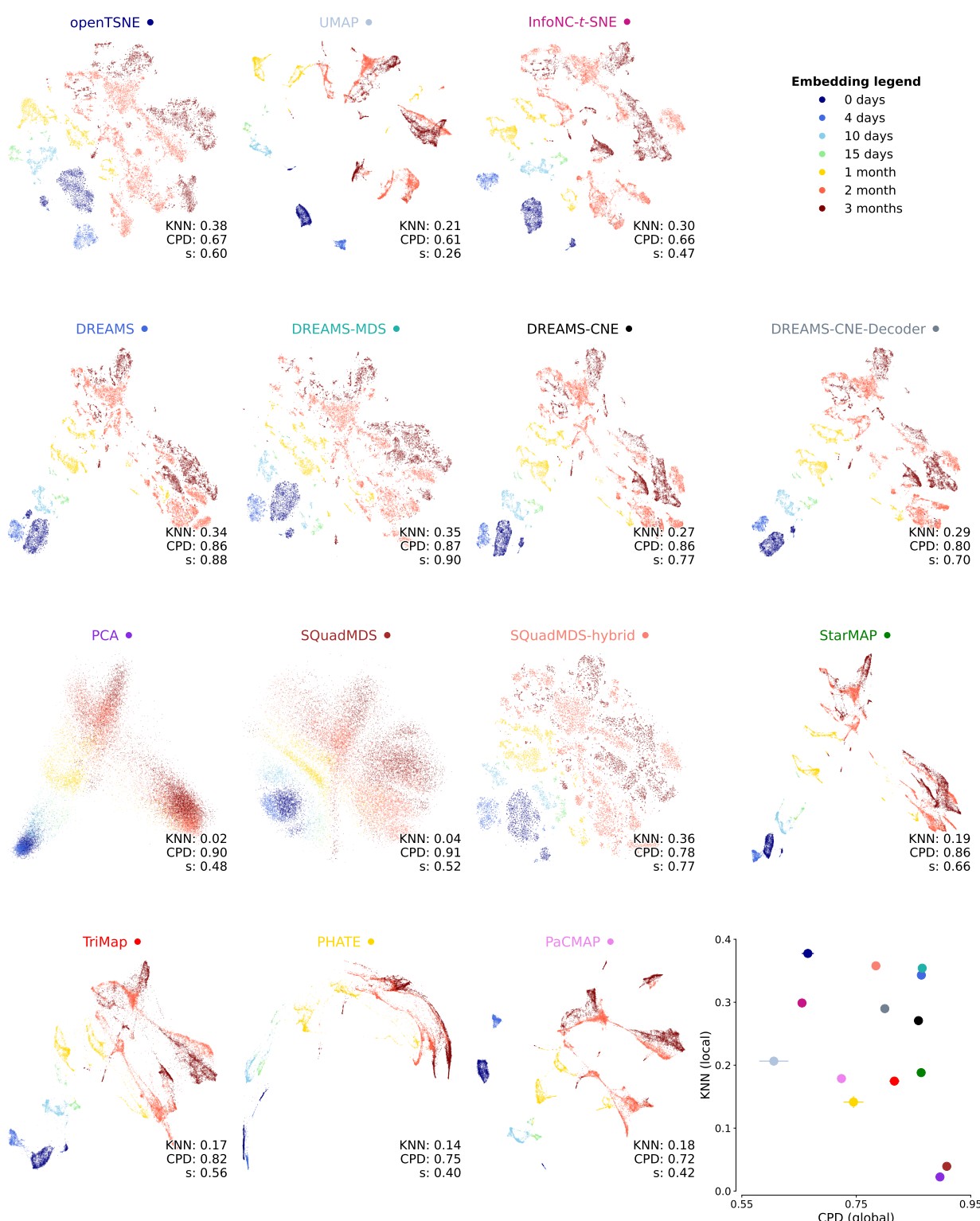

Figure S8: Visualizations of the Kanton et al. dataset with all considered methods.

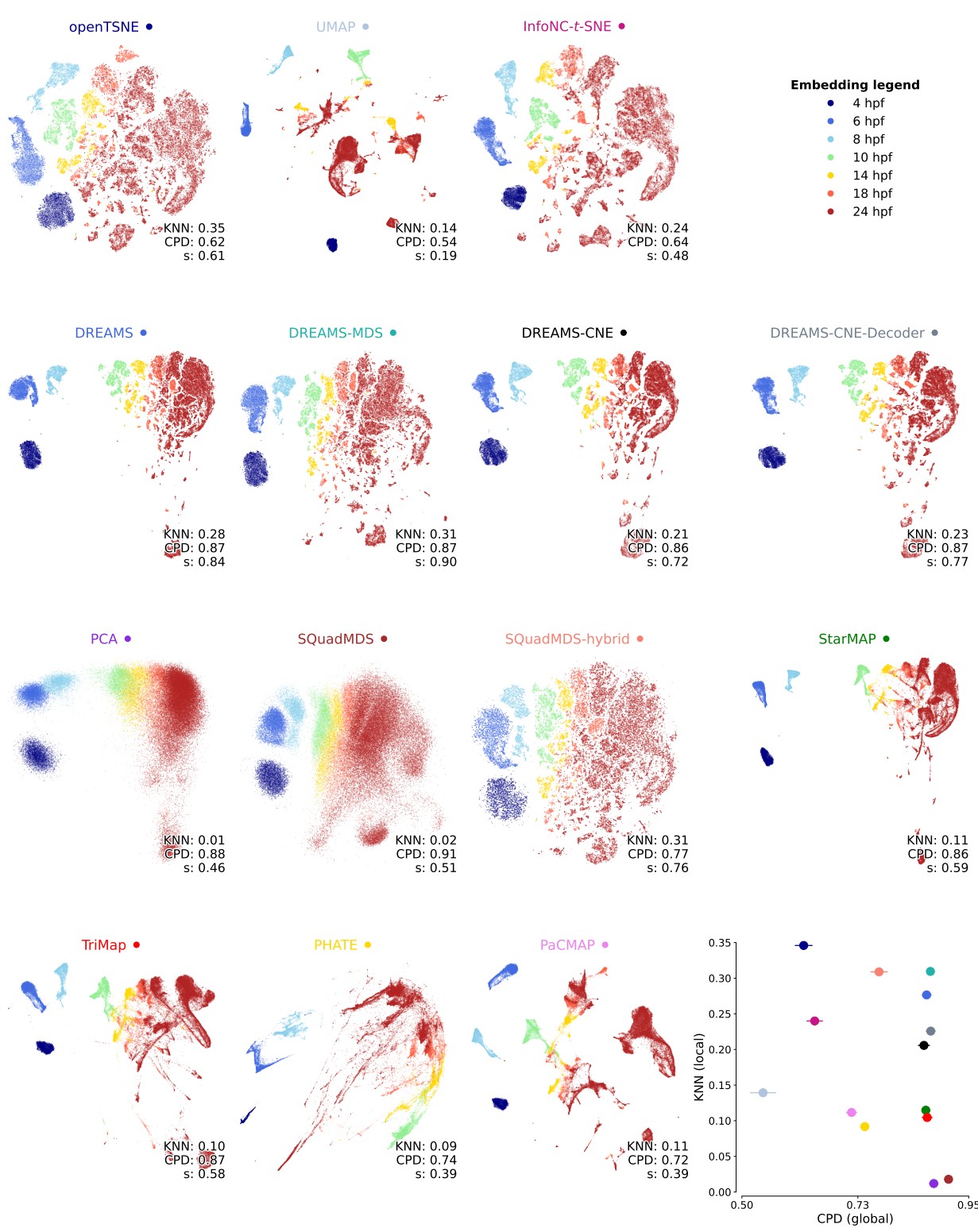

Figure S9: Visualizations of the Wagner et al. dataset with all considered methods (hpf = hours post fertilization).

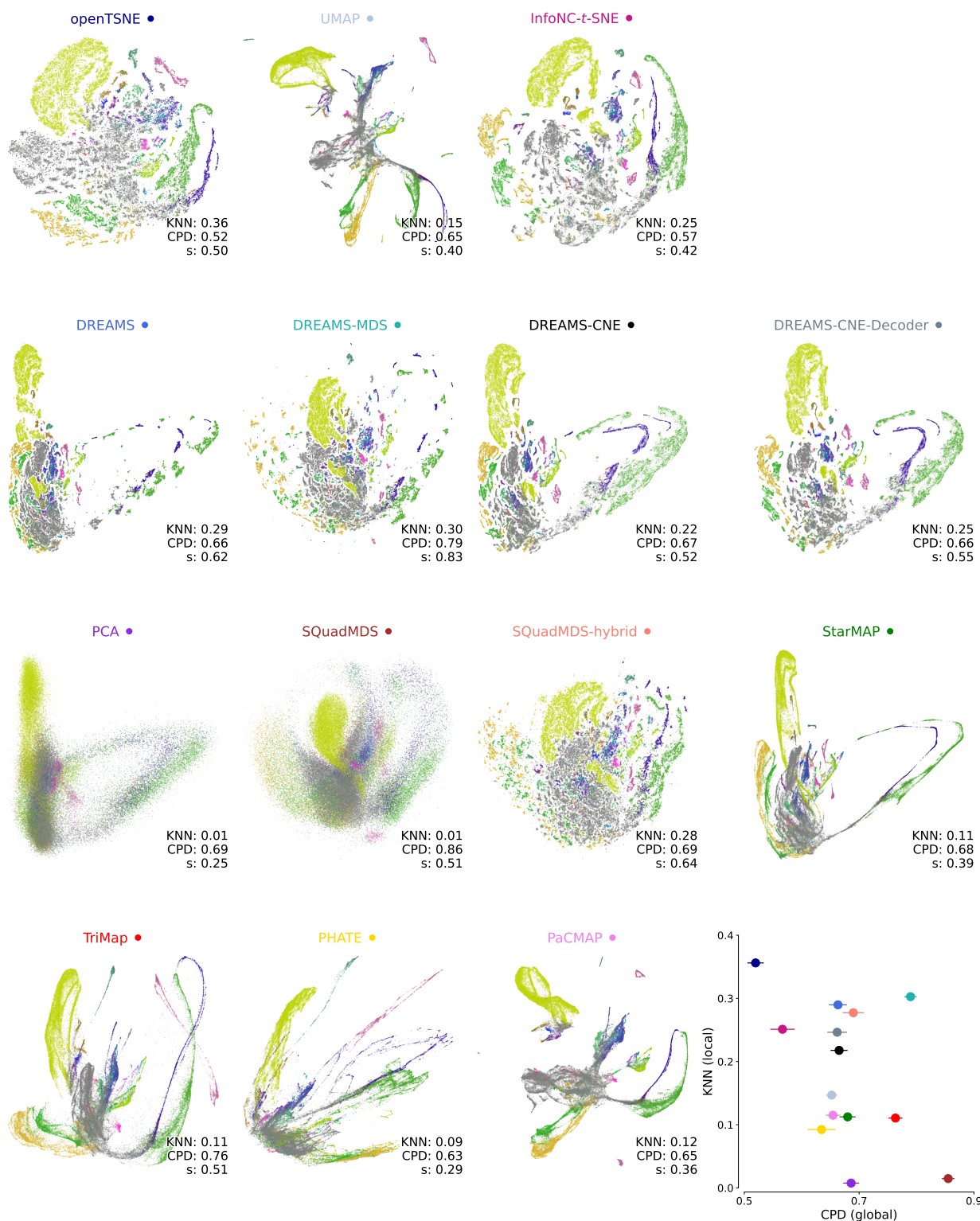

Figure S10: Visualizations of the Packer et al. dataset with all considered methods.

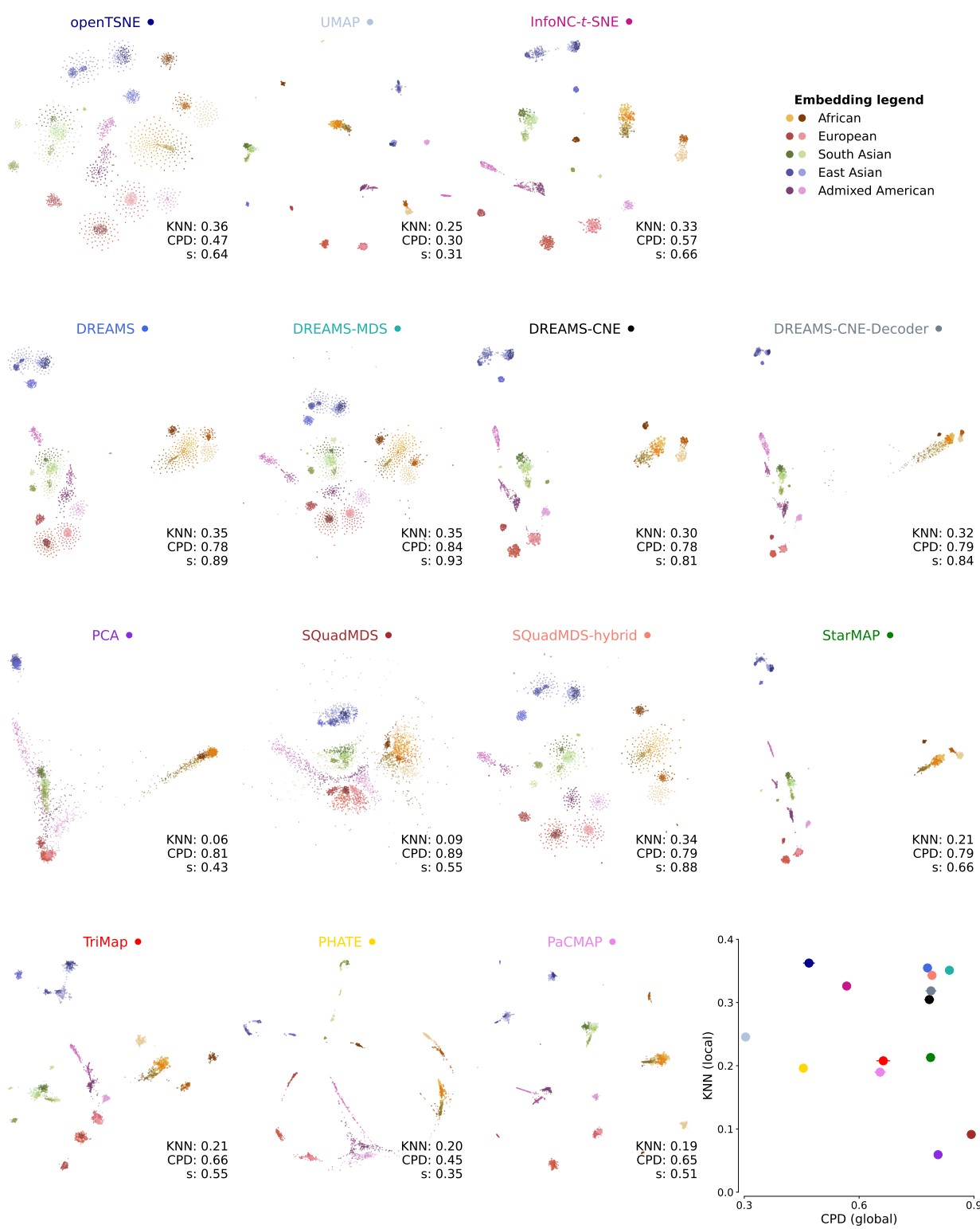

Figure S11: Visualizations of the 1000 Genomes Project dataset (The 1000 Genomes Project Consortium, 2015) with all considered methods.

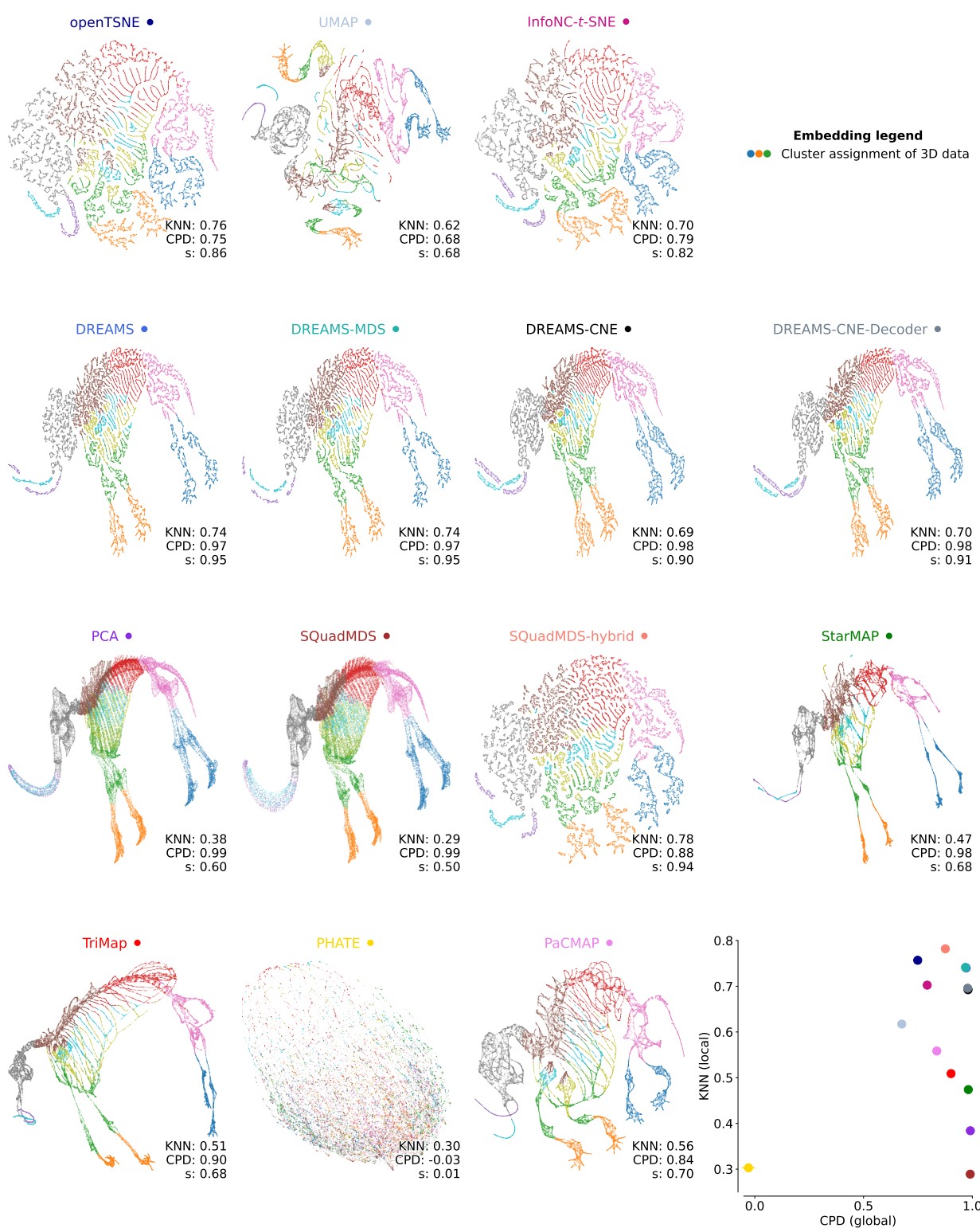

Figure S12: Visualizations of the Mammoth dataset (Smithsonian Institution, 2020; Noichl, 2025) with all considered methods. Colors represent cluster assignment of original data and are solely used for visual clarity but do not correspond to any ground truth labels.

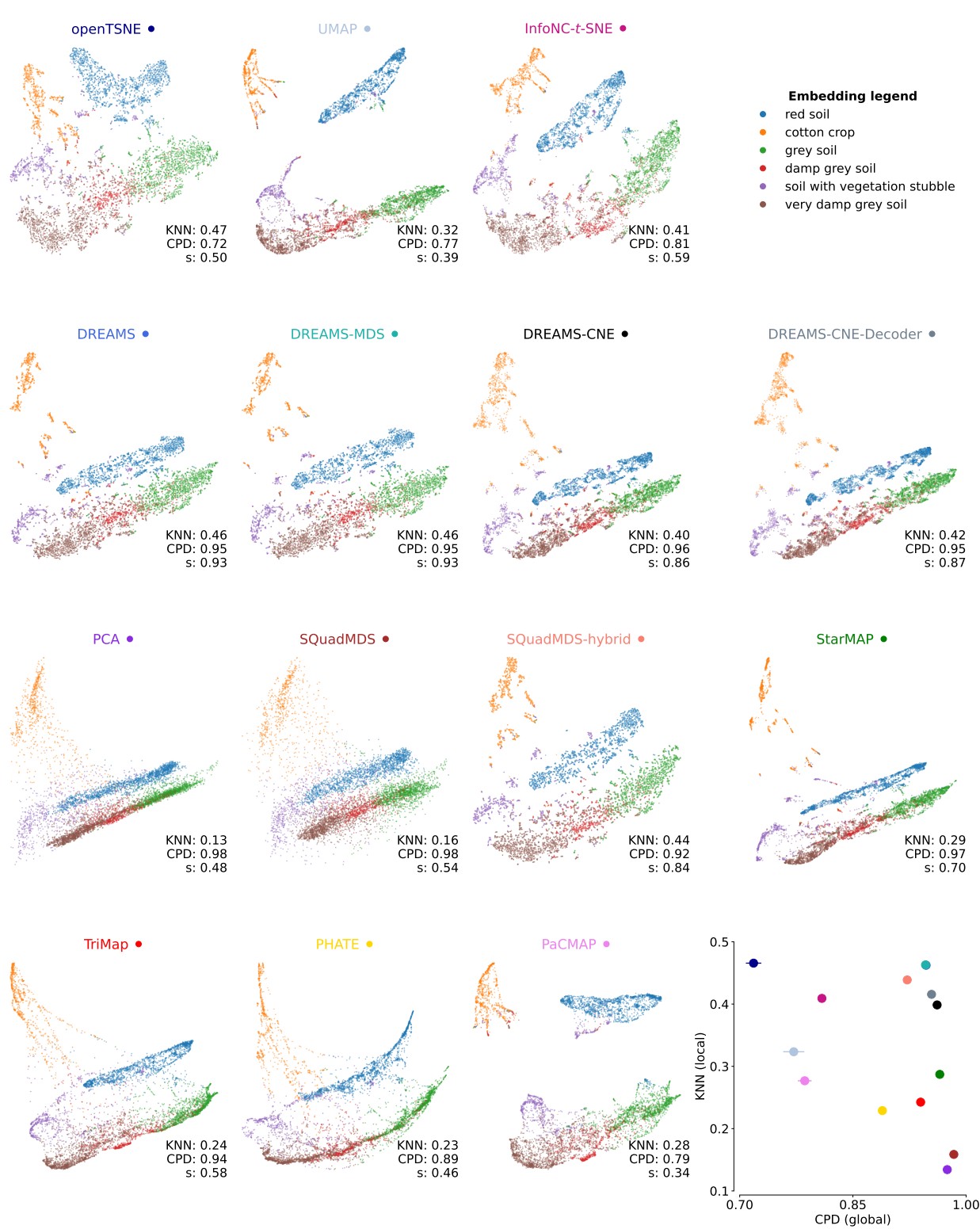

Figure S13: Visualizations of the Landsat Satellite dataset (Srinivasan, 1993) with all considered methods.

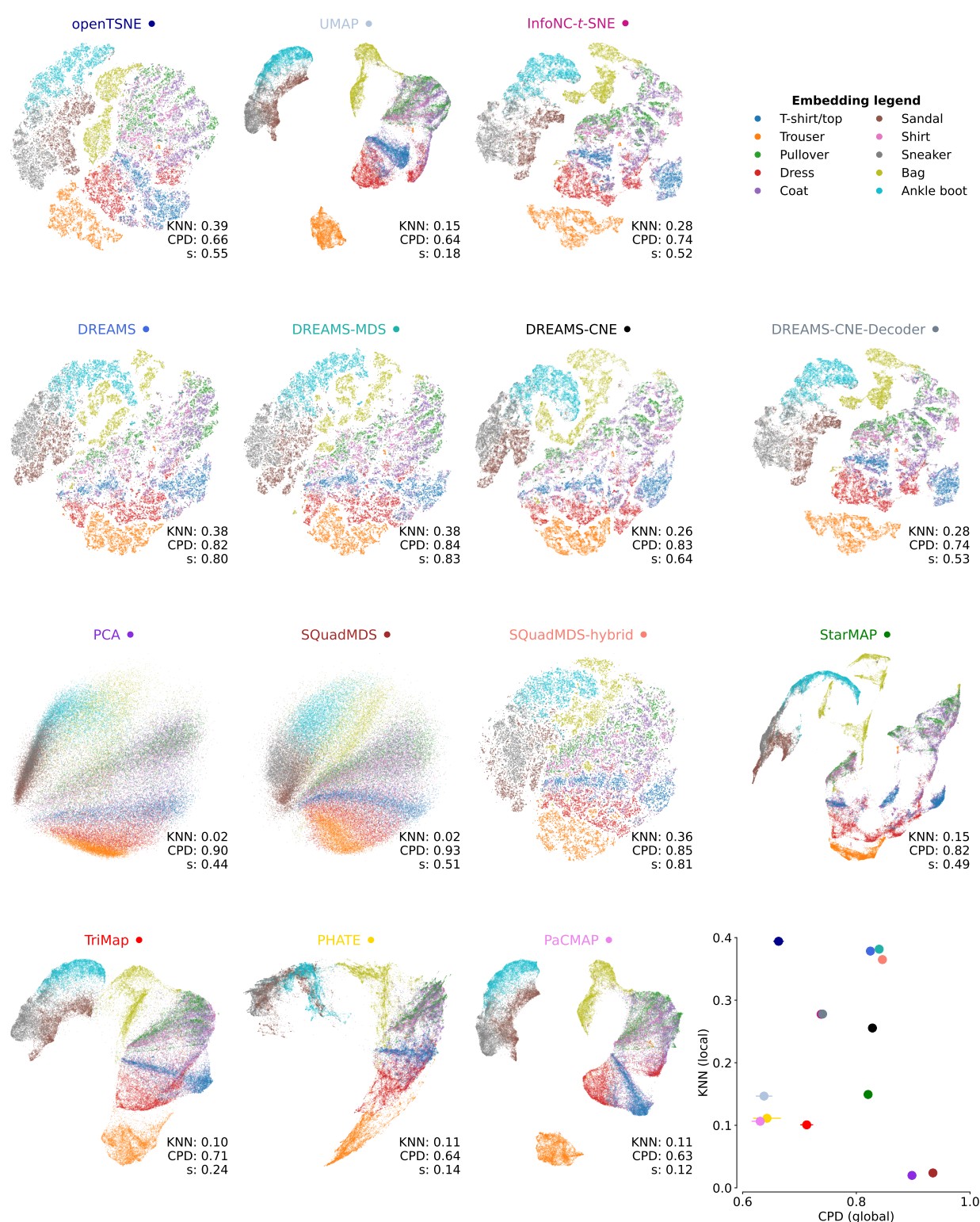

Figure S14: Visualizations of the Fashion MNIST dataset (Xiao et al., 2017) with all considered methods.

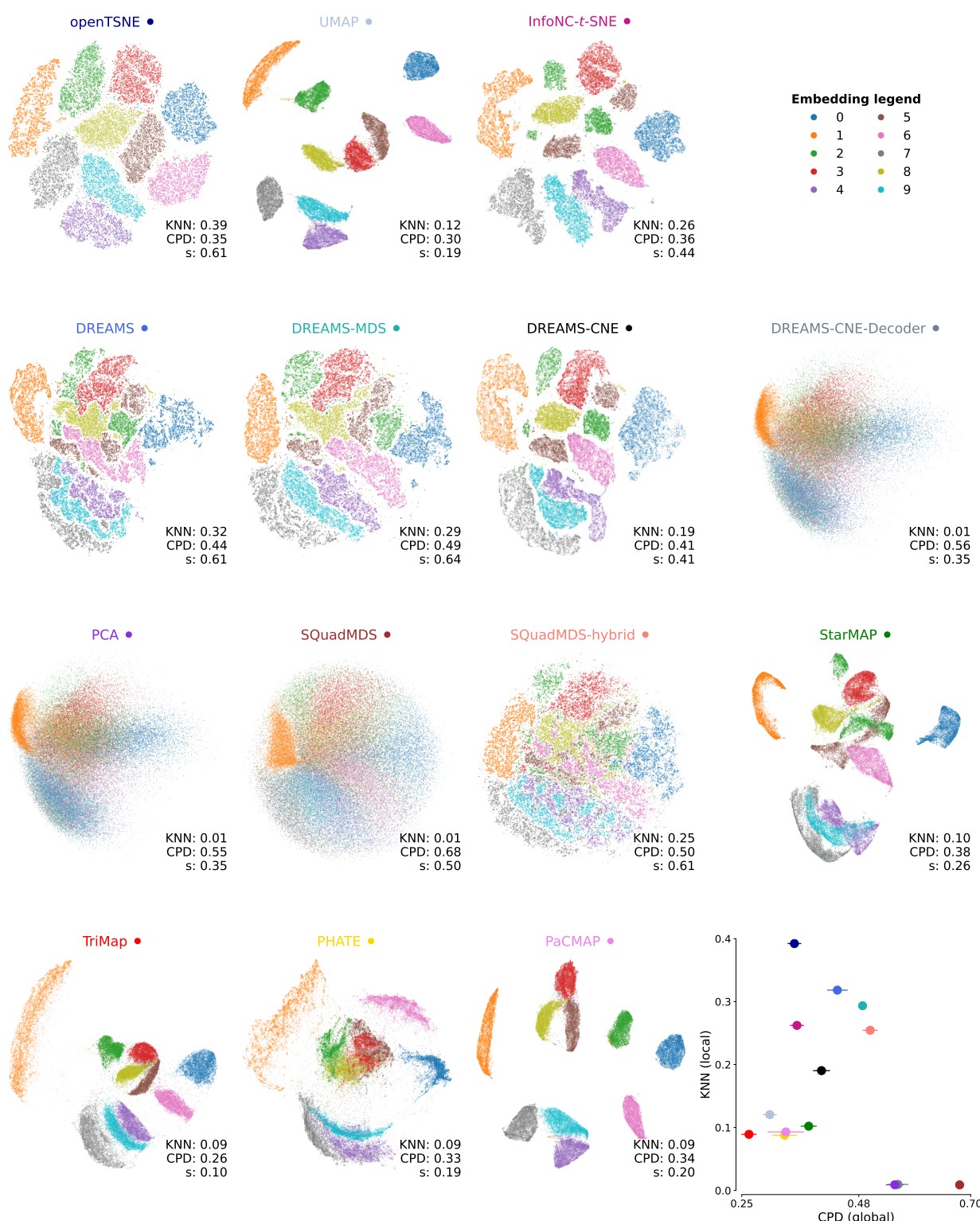

Figure S15: Visualizations of the MNIST dataset (Lecun et al., 1998) with all considered methods.

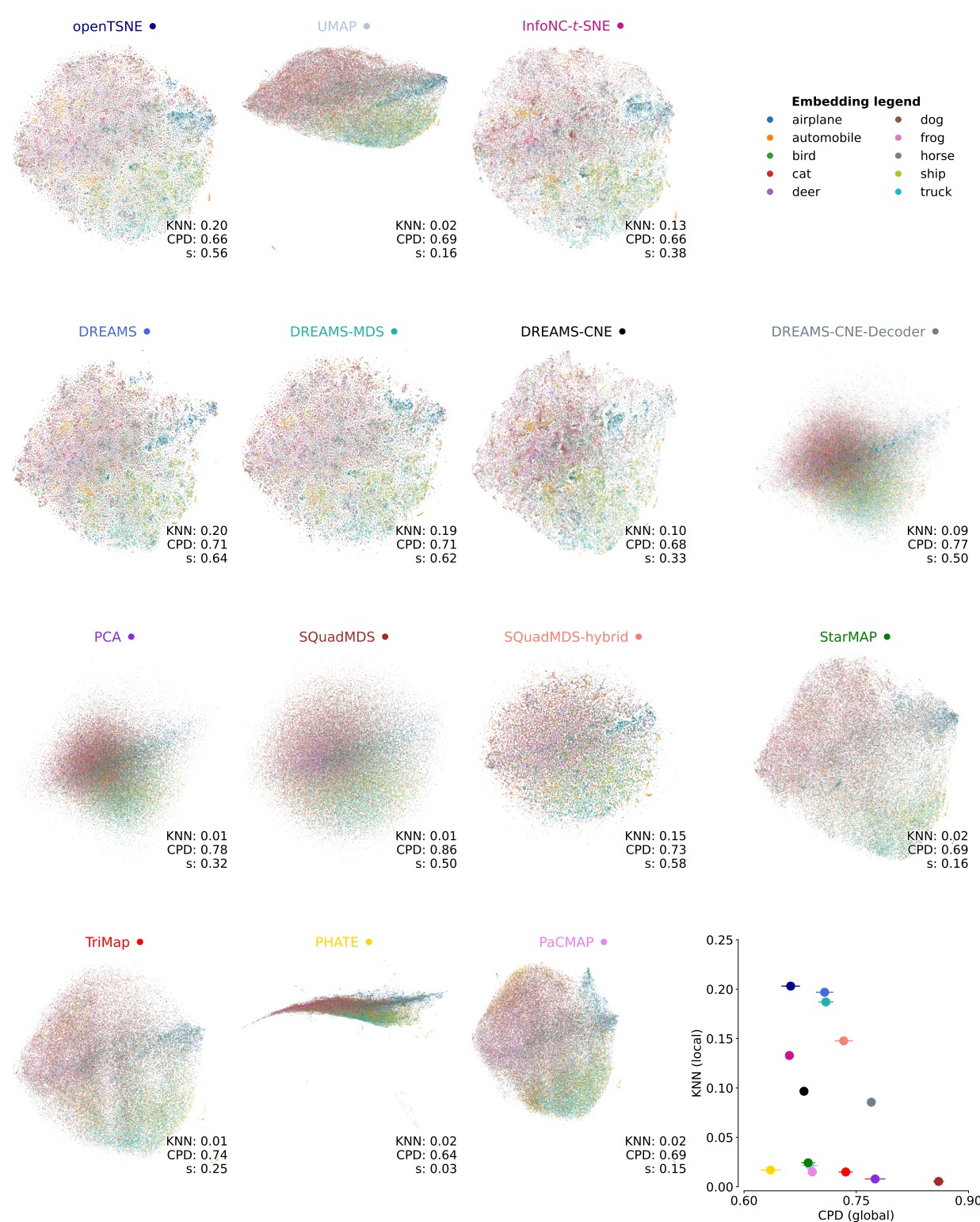

Figure S16: Visualizations of the CIFAR10 dataset (Krizhevsky et al., 2009) with all considered methods.

