# OpenReview forum: "DREAMS: Preserving both Local and Global Structure in Dimensionality Reduction"
_TMLR — Accepted by TMLR_

### Review · Reviewer_yrhb · 2025-09-18

**Summary Of Contributions:**

## Summary
This paper presents DREAMS (Dimensionality Reduction Enhanced Across Multiple Scales), a novel dimensionality reduction method designed to bridge the gap between local and global structure preservation. To achieve this, DREAMS introduces a straightforward regularization term that fuses the strong local structure retention of t-SNE with the robust global structure preservation of PCA. Specifically, this term penalizes deviations of low-dimensional embeddings from precomputed PCA positions—effectively addressing the long-standing trade-off in traditional methods, where tools like t-SNE excel at local structure but struggle with global patterns, and PCA prioritizes global structure at the cost of local detail. The authors conduct rigorous evaluations of 10 dimensionality reduction algorithms across 7 real-world datasets, providing compelling evidence for DREAMS’ effectiveness in balancing local and global preservation.


## Strengths
1. DREAMS outperforms competing methods by effectively retaining both local and global structure. Notably, on most datasets, it achieves KNN scores (a local structure metric) comparable to t-SNE, while its CPD scores (a global structure metric) remain close to those of PCA—demonstrating a rare ability to excel at both scales.
2. DREAMS is not limited to PCA as a global reference; it can integrate other global structure methods (e.g., MDS) to generate variants like DREAMS-MDS. This adaptability makes it well-suited for datasets where MDS (rather than PCA) more accurately captures global patterns, expanding its practical utility.


## Weaknesses
1. While DREAMS is simple and effective, its core innovation is limited: it primarily combines the use of PCA and t-SNE rather than introducing a fundamentally new methodological framework. Additionally, its two-step workflow—first preforming PCA, then optimizing t-SNE with the regularization term—may introduce higher computational costs compared to one-step methods.
2. The regularization term anchors DREAMS’ embeddings to a precomputed global reference (e.g., PCA or MDS). Consequently, if the reference embedding has inherent biases or limitations (e.g., PCA’s tendency to distort non-linear global patterns), these flaws directly propagate to DREAMS’ final output, undermining its performance in such cases.

**Audience:**

Yes

**Audience Explanation:**

Yes, at least some of TMLR’s audience will be interested in this paper’s findings. TMLR focuses on machine learning research, and the paper addresses a long-standing challenge in dimensionality reduction—balancing local and global structure preservation—which is critical for tasks like high-dimensional data visualization (e.g., single-cell transcriptomics, genetic data) that many readers (e.g., researchers in ML for biology, data visualization, or applied ML) work on.

**Broader Impact Concerns:**

No ethical concerns are noted.

**Claims And Evidence:**

Yes

**Claims Explanation:**

Yes, the claims in the submission are supported by accurate, convincing, and clear evidence. The paper validates DREAMS’ ability to balance t-SNE’s local and PCA’s global structure preservation through both qualitative (embedding visualizations across 7 datasets, e.g., Tasic et al.’s clear separation of cell classes and fine clusters) and quantitative metrics (KNN for local, CPD for global, and aggregated scores), where DREAMS or DREAMS-MDS consistently outperforms 10 competing methods. It further confirms flexibility (supporting MDS as a global reference) and reproducibility (detailed experimental setups, open-source implementations, and results averaged across 4 random seeds), ensuring the evidence is rigorous and credible.

**Requested Changes:**

The paper does not evaluate DREAMS on datasets where neither PCA nor t-SNE performs effectively (e.g., highly non-linear datasets with complex, multi-scale structures). It would be valuable to assess how DREAMS performs in these challenging scenarios, as this would clarify its robustness beyond cases where its reference methods already work well.

---

> ### Author Response · Authors · 2025-10-27
>
> Dear reviewer yrhb,
>
> we thank you for the time and effort invested into reviewing our paper.
>
> > core innovation is limited
>
> We acknowledge that DREAMS does not introduce a fundamentally new methodological framework. However, we do not consider this a limitation. DREAMS deliberately builds on the complementary strengths of a global method (PCA or MDS) and a neighbor embedding (t-SNE) in a simple yet effective way. This design enables strong performance while remaining interpretable, easy to implement, and broadly applicable. We consider the method's simplicity in combination with its superior results compared to more complex alternatives an advantage rather than a limitation.
>
> > its two-step workflow—first preforming PCA, then optimizing t-SNE with the regularization term—may introduce higher computational costs compared to one-step methods.
>
> We thank you for pointing out the lack of reported runtimes in the orignal submission. In standard t-SNE the PCA step is typically performed anyway for initialization of the embedding and takes negliglble time compared to t-SNE. Therefore, computing the PCA embedding does not add extra computational cost.
>
> That said, DREAMS' regularizer does add some computational overhead, but it is only around 25% slower than t-SNE. We have clarified this point in the revised Section 5 and included figure S3 comparing runtimes across methods, showing that DREAMS' runtime is comparable t-SNE's.
>
> > if the reference embedding has inherent biases or limitations (e.g., PCA’s tendency to distort non-linear global patterns), these flaws directly propagate to DREAMS’ final output
>
> We agree that biases of the global embedding can propagate into the final embedding and therefore limit its performance. We already explicitly acknowledged and discussed this point in the Discussion (Section 7), noting that this dependency is inherent to our approach. However, DREAMS can incorporate any global embedding, as evidenced by the examples where the non-linear MDS provided a better reference for DREAMS than PCA. In addition, in practice we observe that DREAMS still outperforms competing methods in preserving local and global structure, even in cases where the the reference embedding is suboptimal. For example in the MNIST dataset, where no clear global structure is present (CPD of PCA only at ~0.5), DREAMS still provides a better simultaneous global and local structure preservation than other methods (see Figure 3j and S15).
>
> > The paper does not evaluate DREAMS on datasets where neither PCA nor t-SNE performs effectively
>
> We agree that showcasing challenging cases is important. But we disagree that we did not include such examples. On the Macosko dataset, the kNN recall of t-SNE is only at 0.2. Nevertheless, DREAMS manages to combine this local performance with the good global PCA layout. We explicitly discussed the MNIST dataset as a challenging case as it does not contain a prominent global structure. Here, DREAMS was not able to combine the best possible kNN recall with the best possible CPD but incurred a clear trade-off. Nevertheless, DREAMS achieved the best trade-off between local and global structure among all methods.
>
> As an additional challenging example, we now included the CIFAR10 dataset in the revised version (see figure 3k and S16, where neither t-SNE nor PCA/MDS provides a well-structured embedding. Yet our main conclusion remains valid: t-SNE best preserves local structure, PCA/MDS best preserves global structure, and DREAMS most effectively combines both strengths. Moreover, the simple framework allows to choose any global reference embedding, which makes it adaptable to different datasets and intended uses.

---

### Review · Reviewer_zedR · 2025-09-26

**Summary Of Contributions:**

This paper proposes a new method called DREAMS to simultaneously preserve both “local structure (e.g., neighborhood relationships)” and “global structure (e.g., the overall arrangement of classes)” in the two-dimensional visualization of high-dimensional data—a classic challenge. This is achieved by adding a regularization term derived from PCA to the objective function of t-SNE. Specifically, it introduces weighted regularization that penalizes the squared sum of the difference between the embedding and a pre-obtained global reference embedding (primarily PCA, or MDS in some cases) against the KL loss of t-SNE, aiming to minimize L.

Strengths
- The objective function extension is simple and easy to implement. The ability to obtain a continuous local-global spectrum via lambda is practically useful.
- The experimental design, encompassing not only PCA but also DREAMS-MDS using MDS as a reference, demonstrates the possibility of selecting a “good global” spectrum tailored to data characteristics.
- By placing indicators (KNN/CPD) on both axes and visualizing the trade-offs of each method in Figures 3 and 4, the differences are easily conveyed to readers.

Weaknesses
- Theoretical implications are shallow: The treatment of alpha, the equivalence of solutions when lambda (perfect PCA recovery), and its inverse remain at the level of empirical rules. It would be better to state that they are mathematically based and strictly consistent.

**Additional Comments:**

N/A

**Audience:**

Yes

**Audience Explanation:**

TMLR readers include many practitioners and researchers in dimensionality reduction for representation learning and data visualization, where balancing local and global properties remains a longstanding practical challenge. Findings showing superior trade-offs over existing hybrids with simple regularization are noteworthy.

**Claims And Evidence:**

Yes

**Claims Explanation:**

The systematic visualization and quantitative comparison across seven datasets largely supports the main claim that the proposed method maintains both local and global properties well. The direct comparison of competing methods' spectra via lambda sweep curves is also persuasive. While experimentally promising, theoretical grounding is necessary.

**Requested Changes:**

- Explicitly describe mathematically how PCA and t-SNE are each restored depending on lambda.
- Clarify the treatment of alpha and geometry. State the motivation for fixing α at each iteration and provide stability analysis (presence/absence of contribution to gradients). Present comparisons (at least ablation studies) with introducing Procrustean alignment (scale + orthogonal rotation) into regularization, including rotational degrees of freedom. The current approach uses scale only and may not be invariant to rotation.

---

> ### Author Response · Authors · 2025-10-27
>
> Dear reviewer zedR,
>
> we thank you for the time and effort invested into reviewing our paper.
>
> > Explicitly describe mathematically how PCA and t-SNE are each restored depending on lambda.
>
> For $\lambda = 0$, DREAMS provably produces a standard $t$-SNE embedding since DREAMS' loss term Eq (2) simply reduces to the $t$-SNE objective
> $$
> \mathcal{L}(Y) = \mathcal{L}_{t-\text{SNE}} (Y).
> $$
> As we follow all the defaults of and use the openTSNE backend, this is literally just an instance of t-SNE. In the other limiting case of regularization strength $\lambda= 1$, the objective becomes the regularizer without the $t$-SNE loss
> $$
> \mathcal{L}(Y) = \frac{1}{n}|| Y -  \frac{||Y||_F}{|| \tilde{Y}||_F} \tilde{Y}||_F^2.
> $$
> Its optimal value is zero and achieved precisely by the layouts $Y$, with $\frac{Y}{||Y||_F} = \frac{\tilde Y}{|| \tilde Y ||_F}$, i.e., the layouts $\{\beta \tilde{Y} | \beta > 0\}$, i.e., positively scaled versions of the global reference embedding $\tilde{Y}$.
>
> We summarized this explanation in our revised methods section to clearly demonstrate that the limiting cases of regularization strength $\lambda=0$ and $\lambda=1$ mathematically correspond to the $t$-SNE and PCA (or MDS) embeddings repectively.
>
> >Present comparisons (at least ablation studies) with introducing Procrustean alignment (scale + orthogonal rotation) into regularization, including rotational degrees of freedom. The current approach uses scale only and may not be invariant to rotation.
>
> Thank you for pointing out the possible advantages of using Procrustes alignment. In the beginning of the development of DREAMS, we did consider Procrustes analysis and have its scaling already implemented as a second scaling option (referred to as Procrustes scaling). Instead of our $\alpha = \frac{||Y||_F}{||\tilde{Y}||_F}$, Procrustes uses $\alpha_p = \frac{\langle Y, \tilde{Y} \rangle}{||\tilde{Y}||_F^2}$, which is the scaling that minimizes the regularization loss for given $Y$ and $\tilde{Y}$. By the Cauchy--Schwarz inequality, our $\alpha$ is at least as large as $\alpha_p$ and equals it iff $Y$ is a positive rescaling of $\tilde{Y}$.
>
> However, empirically, scaling with $\alpha_p$ performed slightly worse than the scaling with our $\alpha$ on the Tasic data. While the regularization loss is, by construction, lower with $\alpha_p$, the KL divergence is slightly higher, the scale of the final embedding is further from t-SNE's, and both performance metrics were worse than for our $\alpha$ (see new Figure S2 and new Table S1 (reproduced below)).
>
> The main purpose of scaling $\tilde{Y}$ is to avoid imposing an artificial scale onto the scale-sensitive $t$-SNE embedding. Therefore, the better performance of our larger scaling factor might be due the fact that the resulting embedding scale is closer in to that of a normal t-SNE embedding than with the smaller scaling factor $\alpha_p$.
>
> Your questions led us to discuss the $\alpha_p$ scaling in a new appendix C where we also demonstrate the advantages of our norm-based scaling compared to the Procrustes scaling and the case without any scaling.
>
> Regarding the rotations: it is sufficient to transform $\tilde{Y}$ only by scaling, since the $t$-SNE objective is invariant to translations and rotations (as it only depends on the distances between points) and the $t$-SNE gradients are equivariant to rotations and invariant to translations. Therefore, translating and a rotating the global reference to align it to the DREAMS embedding (i.e., performing full Procrustes alignment), only changes the final embedding by a translation and rotation, which do not influence the visualization quality, but increases computational complexity. See the new Table S1, which we also reproduce here for convenience
>
> | Metric | DREAMS ($\alpha$) | scaling by $\alpha_P$ | $\alpha_P$ + translation | full Procrustes ($\alpha_P$) | no scaling |
> | --- | :---: | :---: | :---: | :---: | :---: |
> |KNN | **0.40** | 0.39 | 0.39 | 0.39 | 0.32 |
> |CPD | **0.89** | 0.86 | 0.86 | 0.86 | 0.53 |

---

> > ### Author Response · Authors · 2025-10-27
> >
> > > State the motivation for fixing $\alpha$ at each iteration and provide stability analysis
> >
> > Let us consider the case where we do compute the gradient of $\alpha$. As discussed above, for given $Y$ and $\tilde{Y}$, the optimal scaling would be $\alpha_p \leq \alpha$. This means that to minimize the regularizer, $\alpha$ should decrease to $\alpha_p$, i.e., $||Y||_F$ needs to decrease. This corresponds to a shrinkage of $Y$. More formally, if $\alpha$ is not treated as a constant, the gradient of the regularizer by $Y$ becomes
> >
> > $$
> > 2\bigg (Y - \frac{||Y||_F}{||\tilde{Y}||_F} \tilde{Y} \bigg ) - 2 \bigg (\frac{\langle Y, \tilde{Y}\rangle}{||Y||_F ||\tilde{Y}||_F} -1 \bigg) Y.
> > $$
> > The first term corresponds to $Y$ becoming similar to $\frac{||Y||_F}{||\tilde{Y}||_F} \tilde{Y}$, the normal gradient of the regularizer if $\alpha$ were a constant. The second term is the contribution of $\alpha$ to the gradient. Since $\frac{\langle Y, \tilde{Y}\rangle_F}{||Y||_F ||\tilde{Y}||_F} \leq 1$, this term corresponds to a shrinkage of $Y$ towards to origin.
> >
> > As mentioned above, since the t-SNE loss is scale-sensitive, we do not want the regularizer to have a bearing on the scale of $Y$. Thus, the shrinkage term is similarly undesirable as using a constant $\alpha_p$ for scaling.
> >
> > Nevertheless, in the CNE variant of DREAMS, we do propagate gradient through $\alpha$ and still obtain a good trade-off between the InfoNC-t-SNE and PCA performance (Fig 4c). We also ran some experiments treating $\alpha$ as a constant in the CNE version (and propagating the gradient through $\alpha$ in the default DREAMS version) and did not observe any performance difference. So the shrinkage when not treating $\alpha$ as a constant seems less drastic and less harmful than using a fixed $\alpha_p$.
> >
> > An additional small advantage of treating $\alpha$ as a constant is that the gradient becomes simpler.
> >
> > We added this discussion as a new appendix C.

---

### Review · Reviewer_gRWA · 2025-10-17

**Summary Of Contributions:**

**Summary**

The paper introduces DREAMS, a dimensionality reduction approach that augments the t-SNE objective with a regularization term penalizing deviations from PCA positions. By encouraging the embedding to PCA, the method aims to preserve global structure during optimization, addressing a common limitation of conventional techniques that tend to emphasize either local or global structure but not both.

**Strengths**

1. The paper is well written and has a complete framework.
2. The proposed algorithm has been applied to seven real-world datasets with multiple baselines.
3. The proposed algorithm outperforms the baselines in terms of the trade-off between preserving local and global structure.
4. The visualization is clear, and the plots are informative.

**Weakness**

1. The theoretical guarantee is unclear. Please refer to the comments in the next section.
2. Most of the chosen datasets focus on a specific area, i.e., scRNA-seq, and I am unsure whether diversity of datasets is needed for evaluation in this area.

**Audience:**

Yes

**Audience Explanation:**

Given the competitive performance on seven real-world datasets relative to other baselines, I think the proposed algorithm will be of interest. However, as mentioned above, the diversity of datasets, the evaluation metrics, and the theoretical component warrant further discussion.

**Claims And Evidence:**

No

**Claims Explanation:**

Empirically, the proposed algorithm appears to balance local and global structure well, and the embedding visualizations look compelling. I have a couple of clarification questions.

Regarding the aggregated local–global score: is this score commonly used in the literature? If not, a further justification of its design and interpretation would be helpful. Also, is the KNN metric you use different from the “KNN Accuracy” in the paper Watanabe et al. 2025 about STARMAP? And is the evaluation sensitive to the choice of $k$?

For the theory, could you clarify how $\alpha$ operates at each iteration and why it can be treated as a constant? When minimizing the modified objective function, is there a theoretical guarantee that both local and global structures are preserved in a balanced manner governed by $\lambda$?

**Requested Changes:**

1. Please clarify the points mentioned above.
2. It would be interesting to see results on other commonly used datasets for dimensionality reduction.

---

> ### Author Response · Authors · 2025-10-27
>
> Dear reviewer gRWA,
>
> we thank you for the time and effort invested into reviewing our paper.
>
> > is there a theoretical guarantee that both local and global structures are preserved in a balanced manner governed by lambda?
>
> No, there is no such theoretical guarantee. While we observe excellent preservation of local and global structure on the datasets with multiple scales, we also observed with the negative examples, MNIST and the newly added CIFAR10, that DREAMS does not always manage to simultaneuously maintain t-SNE's KNN recall and PCA's CPD for any $\lambda$. Moreover, we are doubtful if it is even possible for every dataset to maintain the KNN level of t-SNE and the CPD level of PCA by *any* 2D layout.
> We caution about the lack of theoretical performance guarantees in the revised discussion.
>
> > clarify how alpha operates at each iteration and why it can be treated as a constant?
>
> Before each iteration, we compute the norm $|| Y ||_F$ of the current DREAMS embedding and compute $\alpha = || Y ||_F / || \tilde{Y} ||_F$ (the term $|| \tilde{Y} ||_F$ does not change during training). Since our backend openTSNE does not use automatic differentiation, we can simply insert this $\alpha$ value into the gradient of the regularizer.
>
> We scale the reference embedding, since t-SNE is scale-sensitive. In order to not impose any arbitrary embedding scale through the regularizer, we rescale the global reference embedding to be of comparable scale as the DREAMS embedding and thus close to the scale that the t-SNE loss would produce. This makes the regularizer more compatible with the t-SNE loss and signficantly improves the final loss values and the visualization quality. Since the embedding scale does not influence the visual quality, we can choose it as needed and opted for a constant $\alpha$ mainly for simplicity.
>
> Treating $\alpha$ not as a constant, but propagating the gradient through it, also works, as evidenced by our CNE variant. For the revision, we added a new appendix C detailing alternative choices for $\alpha$ and the effect of keeping it a constant.
>
>
> > diversity of datasets
>
> We selected the seven presented datasets based on their suitability for evaluating both local and global structure preservation. ScRNA-seq data often exhibits meaningful structure at both local and global scale, making them particularly well suited for our method.
>
> Moreover, data visualization is widely used in the two domains scRNA-seq (Kobak & Berens, 2019) and population genomics (Diaz-Papkovich et al., 2020). To further demonstrate the effectiveness of DREAMS outside of the scRNA-seq domain, we extended the set of non-scRNA-seq datasets beyond 1000 Genomes and MNIST by four additional datasets common in dimensionality reducion (Mammoth, Fashion MNIST, Sattellite, CIFAR10) and comprising both image and 3D point cloud data. We found that on Mammoth, Fashion MNIST, and Sattellite, DREAMS was again able to obtain near perfect combinations of t-SNE's good local and PCA's good global performance outperforming all other methods. For the CIFAR10 dataset, we deliberately used the raw pixel values as input to make this dataset challenging. Indeed, the visulisations by all methods were poor. Nevertheless our main conclusion remained valid: t-SNE best preserved local structure, PCA/MDS best preserved global structure, and DREAMS most effectively combined both strengths. However, DREAMS incured a clear trade-off and could not preserve local and global structure simultaneously.
>
> > aggregated local–global score: is this score commonly used in the literature? If not, a further justification of its design and interpretation would be helpful
>
> Our local-global score is not a standard metric. We introduced it to aggregate our two main and well-established metrics (KNN and CPD) in one score. In the revision, we complemented its description after Eq (5) by a new figure S4 visualizing the metric. The local-global score is between 0 and 1 with 1 being the best. A method achieves score 1 only if it achieves both the best local (KNN) and the best global (CPD) score and conversely it gets score 0 if it obtains both the worst global and the worst local score. Methods that obtain the best local but worst global (like t-SNE in Fig S4 right) or the best global and worst local score (like PCA in Fig S4 left) achieve a score of 0.5. The score corresponds visually to the depicted panels in Figs 3 and S4, where the top right corner is best and the bottom left corner worst.

---

> > ### Author Response · Authors · 2025-10-27
> >
> > > is the KNN metric you use different from the “KNN Accuracy” in the paper Watanabe et al. 2025 about STARMAP?
> >
> > Yes, our KNN metric is different from Wanatabe et al.'s "KNN Accuracy". Wanatabe et al.'s metric is a kNN *classification accuracy*, which relies on available ground truth class labels. It measures if the k nearest neighbors in the low-dimensional embedding have the same class as the reference point. We use kNN *recall*, a very common metric for dimensionality reduction [a, b]. As we described in Section 5.1, it computes the share of k-nearest neighbors in high-dimensional space that are among the k-nearest neighbors in the low-dimensional embedding. It does not require any class labels. Both metrics focus on the local structure of the embedding. kNN accuracy is usually a coarser metric, as it only considers class membership, while kNN recall considers the exact nearest neighbors (which are typically also of the same class as the reference point).
> >
> > > is the evaluation [of the KNN metric] sensitive to the choice of k?
> >
> > Yes, the kNN recall is indeed sensitive to the choice of k. Following [a], we used a small k=10, as this metric is intended to quantify the local structure quality. Moreover, as k approaches the number of datapoints, this metric naturally approaches 1, which further justifies our choice of a small k. In addition, we added an experiment monitoring the KNN metric and the local-global score for different choices of k. The experiment undermined the good performance of DREAMS across differnt choices of k, producing for all but one of the choices of k the highest local-global score. In the revised version figure S5 has been added to illustrate the new experimental results.
> >
> > [a] Kobak, D., & Berens, P. (2019). The art of using t-SNE for single-cell transcriptomics. Nature communications
> > [b] Lee, J. A., & Verleysen, M. (2009). Quality assessment of dimensionality reduction: Rank-based criteria. Neurocomputing

---

### Decision · Action_Editor_Rt2f · 2025-12-08

**Recommendation:** Accept as is

**Audience:**

Yes

**Audience Explanation:**

The paper proposes a concrete method that is easy to understand and use, and they also provide an open software implementation. The tool allows easy control of the weighting between the PCA and t-SNE extremes and appears robust. Since both PCA and t-SNE are very broadly used as visualization tools, I can well see interest for a tool that allows combining them and enables e.g. fixing some of the pathological properties of either extreme, especially in other sciences where t-SNE are commonly used as general purpose visualizations of structure. In many cases it could be beneficial for them to consider this regularized variant that better aligns with the global structure, here shown to not imply significant drop in preservation of the local structure.

The broader interest within core machine learning community is limited because the methodological development is not particularly deep; the solution is ultimately a relatively straightforward regularization added for t-SNE and there is no theoretical analysis. Nevertheless, the paper still clearly satisfies the evaluation criterion.

**Claims And Evidence:**

Yes

**Claims Explanation:**

The paper proposes a method that combines local and global structure preservation in dimensionality reduction, with a tunable spectrum of embeddings between the two extremes of PCA and t-SNE.

The authors evaluate both properties of the embeddings using established metrics and show in comprehensive evaluation (11 benchmark datasets) that they can indeed form such a continuum. Using a newly designed metric that characterises the overall global-local balance, the authors quantitatively show that the proposed method outperforms a few reasonable comparison methods. While this metric is only one of possibly many that could be used to quantify the overall quality, the results are still clear and confirm the main claims.